# Robust Conditional Probabilities

**Yoav Wald**
School of Computer Science and Engineering
Hebrew University
yoav.wald@mail.huji.ac.il

**Amir Globerson**
The Balvatnik School of Computer Science
Tel-Aviv University
gamir@mail.tau.ac.il

## Abstract

Conditional probabilities are a core concept in machine learning. For example, optimal prediction of a label $Y$ given an input $X$ corresponds to maximizing the conditional probability of $Y$ given $X$. A common approach to inference tasks is learning a model of conditional probabilities. However, these models are often based on strong assumptions (e.g., log-linear models), and hence their estimate of conditional probabilities is not robust and is highly dependent on the validity of their assumptions.

Here we propose a framework for reasoning about conditional probabilities without assuming anything about the underlying distributions, except knowledge of their second order marginals, which can be estimated from data. We show how this setting leads to guaranteed bounds on conditional probabilities, which can be calculated efficiently in a variety of settings, including structured-prediction. Finally, we apply them to semi-supervised deep learning, obtaining results competitive with variational autoencoders.

## 1   Introduction

In classification tasks the goal is to predict a label $Y$ for an object $X$. Assuming that the joint distribution of these two variables is $p^*(\boldsymbol{x}, \boldsymbol{y})$ then optimal prediction[1] corresponds to returning the label $\boldsymbol{y}$ that maximizes the conditional probability $p^*(\boldsymbol{y}|\boldsymbol{x})$. Thus, being able to reason about conditional probabilities is fundamental to machine learning and probabilistic inference.

In the fully supervised setting, one can sidestep the task of estimating conditional probabilities by directly learning a classifier in a discriminative fashion. However, in unsupervised or semi-supervised settings, a reliable estimate of the conditional distributions becomes important. For example, consider a self-training [17, 31] or active learning setting. In both scenarios, the learner has a set of unlabeled samples and it needs to choose which ones to tag. Given an unlabeled sample $\boldsymbol{x}$, if we could reliably conclude that $p^*(\boldsymbol{y}|\boldsymbol{x})$ is close to 1 for some label $\boldsymbol{y}$, we could easily decide whether to tag $\boldsymbol{x}$ or not. Intuitively, an active learner would prefer not to tag $\boldsymbol{x}$ while a self training algorithm would tag it.

There are of course many approaches to "modelling" conditional distributions, from logistic regression to conditional random fields. However, these do not come with any guarantees of approximations to the true underlying conditional distributions of $p^*$ and thus cannot be used to reliably reason about these. This is due to the fact that such models make assumptions about the conditionals (e.g., conditional independence or parametric), which are unlikely to be satisfied in practice.

As an illustrative example for our motivation and setup, consider a set of $n$ binary variables $X_1, ..., X_n$ whose distribution we are interested in. Suppose we have enough data to obtain the joint marginals, $\mathbb{P}[X_i = x_i, X_j = x_j]$, of pairs $i, j$ in a set $E$. If $(1, 2) \in E$ and we concluded that $\mathbb{P}[X_1 = 1 | X_2 = 1] = 1$, this lets us reason about many other probabilities. For example, we know that $\mathbb{P}[X_1 = 1 | X_2 = 1, \ldots, X_n = x_n] = 1$ for *any* setting of the $x_3, \ldots, x_n$

variables. This is a simple but powerful observation, as it translates knowledge about probabilities over small subsets to *robust* estimates of conditional probability over large subsets. Now, what happens when $\mathbb{P}\left[X_1 = 1 | X_2 = 1\right] = 0.99$? In other words, what can we say about $\mathbb{P}\left[X_1 = 1 | X_2 = 1, \ldots, X_n = x_n\right]$ given information about probabilities $\mathbb{P}\left[X_i = x_i, X_j = x_j\right]$. As we show here, it is still possible to reason about such conditional probabilities even under this partial knowledge.

Motivated by the above, we propose a novel model-free approach for reasoning about conditional probabilities. Specifically, we shall show how conditional probabilities can be lower bounded when the only assumption made is that certain low-order marginals of the distribution are known. One of the surprising outcomes of our analysis is that these lower bounds can be calculated efficiently, and often have an elegant closed form. Finally, we show how these bounds can be used in a semi-supervised setting, obtaining results that are competitive with variational autoencoders [11].

## 2 Problem Setup

We begin by defining notations to be used in what follows. Let $X$ denote a vector of random variables $X_1, \ldots, X_n$ which are the features and $Y$ denote labels. If we have a single label we will denote it by $Y$, otherwise, a multivariate label will be denoted by $Y_1, \ldots, Y_r$. $X, Y$ are generated by an unknown underlying distribution $p^*(X, Y)$. All variables are discrete (i.e., can take on a finite set of values).

Here we will assume that although we do not know $p^*$ we have access to some of its low order marginals, such as those of a single feature and a label:

$$\mu_i(x_i, y) = \sum_{\bar{x}_1, \ldots, \bar{x}_n : \bar{x}_i = x_i} p^*(\bar{x}_1, \ldots, \bar{x}_n, y).$$

Similarly we may have access to the set of pairwise marginals $\mu_{ij}(x_i, x_j, y)$ for all $i, j \in E$, where the set $E$ corresponds to edges of a graph $G$ (see also [7]). Denote the set of all such marginals by $\boldsymbol{\mu}$. For simplicity we assume the marginals are exact. Generally they are of course only approximate, but concentration bounds can be used to quantify this accuracy as a function of data size. Furthermore, most of the methods described here can be extended to inexact marginals (e.g., see [6] for an approach that can be applied here).

Since $\boldsymbol{\mu}$ does not uniquely specify a distribution $p^*$, we will be interested in the set of all distributions that attain these marginals. Denote this set by $\mathcal{P}(\boldsymbol{\mu})$, namely:

$$\mathcal{P}(\boldsymbol{\mu}) = \left\{ p \in \Delta : \sum_{\bar{x}_1, \ldots, \bar{x}_n : \bar{x}_i = x_i} p(\bar{x}_1, \ldots, \bar{x}_n, y) = \mu_i(x_i, y) \quad \forall i \right\} \tag{1}$$

where $\Delta$ is the probability simplex of the appropriate dimension.

More generally, one may consider some vector function $\boldsymbol{f} : X, Y \to \mathbb{R}^d$ and its expected value according to $p^*$, denoted by $\boldsymbol{a} = \mathbb{E}_{p^*}\left[\boldsymbol{f}(X, Y)\right]$. Then the corresponding set of distributions is:
$$\mathcal{P}(\boldsymbol{a}) = \left\{ p \in \Delta : \mathbb{E}_p\left[\boldsymbol{f}(X, Y)\right] = \boldsymbol{a} \right\}.$$
Since marginals are expectations of random variables [30], this generalizes the notation given above.

### 2.1 The Robust Conditionals Problem

Our approach is to reason about conditional distributions using only the fact that $p^* \in \mathcal{P}(\boldsymbol{\mu})$. Our key goal is to lower bound these conditionals, since this will allow us to conclude that certain labels are highly likely in cases where the lower bound is large. We shall also be interested in upper and lower bounding joint probabilities, since these will play a key role in bounding the conditionals.

Our goal is thus to solve the following optimization problems.
$$\min_{p \in \mathcal{P}(\boldsymbol{\mu})} p(\boldsymbol{x}, \boldsymbol{y}), \quad \max_{p \in \mathcal{P}(\boldsymbol{\mu})} p(\boldsymbol{x}, \boldsymbol{y}), \quad \min_{p \in \mathcal{P}(\boldsymbol{\mu})} p\left(\boldsymbol{y} \mid \boldsymbol{x}\right). \tag{2}$$
In all three problems, the constraint set is linear in $p$. However, note that $p$ is specified by an exponential number of variables (one per assignment $x_1, \ldots, x_n, \boldsymbol{y}$) and thus it is not feasible to plug these constraints into an LP solver. In terms of objective, the min and max problems are linear, and the conditional is fractional linear. In what follows we show how all three problems can be solved efficiently for tree shaped graphs.

# 3 Related Work

The problem of reasoning about a distribution based on its expected values has a long history, with many beautiful mathematical results. An early example is the classical Chebyshev inequality, which bounds the tail of a distribution given its first and second moments. This was significantly extended in the Chebyshev Markov Stieltjes inequality [2]. More recently, various generalized Chebyshev inequalities have been developed [4, 22, 27] and some further results tying moments with bounds on probabilities have been shown (e.g. [18]). A typical statement of these is that several moments are given, and one seeks the minimum measure of some set $S$ under any distribution that agrees with the moments. As [4] notes, most of these problems are NP hard, with isolated cases of tractability. Such inequalities have been used to obtain minimax optimal linear classifiers in [14]. The moment problems we consider here are very different from those considered previously, in terms of the finite support we require, our focus on bounding probabilities and conditional probabilities of assignments.

The above approaches consider worst case bounds on probabilities of events for distributions in $\mathcal{P}(\boldsymbol{a})$. A different approach is to pick a particular distribution in $\mathcal{P}(\boldsymbol{a})$ as an approximation (or model) of $p^*$. The most common choice here is the maximum entropy distribution in $\mathcal{P}(\boldsymbol{a})$. Such log-linear models have found widespread use in statistics and machine learning. In particular, most graphical models can be viewed as distributions of this type (e.g., see [12, 13]). However, probabilities given by these models cannot be related to the true probabilities in any sense (e.g., upper or lower bound). This is where our approach markedly differs from entropy based assumptions. Another approach to reduce modeling assumptions is robust optimization, where data and certain model parameters are assumed not to be known precisely, and optimality is sought in a worst case adversarial setting. This approach has been applied to machine learning in various settings (e.g, see [32, 16]), establishing close links to regularization. None of these approaches considers bounding probabilities as is our focus here.

Finally, another elegant moment approach is that based on kernel mean embedding [23, 24]. In this approach, one maps a distribution into a set of expected values of a set of functions (possibly infinite). The key observation is that this *mean embedding* lies in an RKHS, and hence many operations can be done implicitly. Most of the applications of this idea assume that the set of functions is rich enough to fully specify the distribution (i.e., *characteristic kernels* [25]). The focus is thus different from ours, where moments are not assumed to be fully informative, and the set $\mathcal{P}(\boldsymbol{a})$ contains many possible distributions. It would however be interesting to study possible uses of RKHS in our setting.

# 4 Calculating Robust Conditional Probabilities

The optimization problems in Eq. (2) are linear programs (LP) and fractional LPs, where the number of variables scales exponentially with $n$. Yet, as we show in this section and Section 5, it turns out that in many non-trivial cases, they can be efficiently solved. Our focus below is on the case where the set of edges $E$ corresponding to the pairwise marginals forms a tree structured graph. The tree structure assumption is common in literature on Graphical Models, only here we do not make an inductive assumption on the generating distribution (i.e., we make none of the conditional independence assumptions that are implied by tree-structured graphical models). In the following sections we study solutions of robust conditional probabilities under the tree assumption. We will also discuss some extensions to the cyclic case. Finally, note that although the derivations here are for pairwise marginals, these can be extended to the non-pairwise case by considering clique-trees [e.g., see 30]. Pairs are used here to allow a clearer presentation.

In what follows, we show that the conditional lower bound has a simple structure as stated in Theorem 4.1. This result does not immediately suggest an efficient algorithm since its denominator includes an exponentially sized LP. Next, in Section 4.2 we show how this LP can be reduced to polynomial sized, resulting in an efficient algorithm for the lower bound. Finally, in Section 5 we show that in certain cases there is no need to use a general purpose LP solver and the problem can be solved either in closed form or via combinatorial algorithms. Detailed proofs are provided in the supplementary file.

## 4.1 From Conditional Probabilities To Maximum Probabilities with Exclusion

The main result of this section will reduce calculation of the robust conditional probability for $p(\boldsymbol{y} \mid \boldsymbol{x})$, to one of maximizing the probability of all labels other than $\boldsymbol{y}$. This reduction by itself will not allow for efficient calculation of the desired conditional probabilities, as the new problem is also

a large LP that needs to be solved. Still the result will take us one step further towards a solution, as it reveals the probability mass a minimizing distribution $p$ will assign to $\boldsymbol{x}, \boldsymbol{y}$.

This part of the solution is related to a result from [8], where the authors derive the solution of $\min_{p \in \mathcal{P}(\boldsymbol{\mu})} p(\boldsymbol{x}, \boldsymbol{y})$. They prove that under the tree assumption this problem has a simple closed form solution, given by the functional $I(\boldsymbol{x}, \boldsymbol{y} ; \boldsymbol{\mu})$:

$$I(\boldsymbol{x}, y ; \boldsymbol{\mu}) = \left[ \sum_i (1 - d_i) \mu_i(x_i, y) + \sum_{ij \in E} \mu_{ij}(x_i, x_j, y) \right]_+ . \tag{3}$$

Here $[\cdot]_+$ denotes the ReLU function $[z]_+ = \max\{z, 0\}$ and $d_i$ is the degree of node $i$ in $G$.

It turns out that robust conditional probabilities will assign the event $\boldsymbol{x}, \boldsymbol{y}$ its minimal possible probability as given in Eq. (3). Moreover, it will assign all other labels their maximum possible probability. This is indeed a behaviour that may be expected from a robust bound, we formalize it in the main result for this part:

**Theorem 4.1** *Let $\boldsymbol{\mu}$ be a vector of tree-structured pairwise marginals, then*

$$\min_{p \in \mathcal{P}(\boldsymbol{\mu})} p(\boldsymbol{y} \mid \boldsymbol{x}) = \frac{I(\boldsymbol{x}, \boldsymbol{y} ; \boldsymbol{\mu})}{I(\boldsymbol{x}, \boldsymbol{y} ; \boldsymbol{\mu}) + \max_{p \in \mathcal{P}(\boldsymbol{\mu})} \sum_{\bar{\boldsymbol{y}} \neq \boldsymbol{y}} p(\boldsymbol{x}, \bar{\boldsymbol{y}})} . \tag{4}$$

The proof of this theorem is rather technical and we leave it for the supplementary material.

We note that the above result also applies to the "structured-prediction" setting where $\boldsymbol{y}$ is multivariate and we also assume knowledge of marginals $\mu(y_i, y_j)$. In this case, the expression for $I(\boldsymbol{x}, \boldsymbol{y} ; \boldsymbol{\mu})$ will also include edges between $y_i$ variables, and incorporate their degrees in the graph.

The important implication of Theorem 4.1 is that it reduces the minimum conditional problem to that of probability maximization with an assignment exclusion. Namely:

$$\max_{p \in \mathcal{P}(\boldsymbol{\mu})} \sum_{\bar{\boldsymbol{y}} \neq \boldsymbol{y}} p(\boldsymbol{x}, \bar{\boldsymbol{y}}). \tag{5}$$

Although this is still a problem with an exponential number of variables, we show in the next section that it can be solved efficiently.

## 4.2 Minimizing and Maximizing Probabilities

To provide an efficient solution for Eq. (5), we turn to a class of joint probability bounding problems. Assume we constrain each variable $X_i$ and $Y_j$ to a subset $\bar{X}_i, \bar{Y}_j$ of its domain and would like to reason about the probability of this constrained set of joint assignments:

$$U = \left\{ \boldsymbol{x}, \boldsymbol{y} \mid x_i \in \bar{X}_i, y_j \in \bar{Y}_j \quad \forall i \in [n], j \in [r] \right\} . \tag{6}$$

Under this setting, an efficient algorithm for solving

$$\max_{p \in \mathcal{P}(\boldsymbol{\mu})} \sum_{\boldsymbol{u} \in U \setminus (\boldsymbol{x}, \boldsymbol{y})} p(\boldsymbol{u}),$$

will also solve Eq. (5). By the results of last section, we will then also have an algorithm calculates robust conditional probabilities. To see this is indeed the case, assume we are given an assignment $(\boldsymbol{x}, \boldsymbol{y})$. Then setting $\bar{X}_i = \{x_i\}$ for all features and $\bar{Y}_j = \{1, \ldots, |Y_j|\}$ for labels (i.e. $U$ does not restrict labels), gives exactly Eq. (5).

To derive the algorithm, we will find a compact representation of the LP, with a polynomial number of variables and constraints. The result is obtained by using tools from the literature on Graphical Models. It shows how to formulate probability maximisation problems over $U$ as problems constrained by the local marginal polytope [30]. Its definition in our setting slightly deviates from its standard definition, as it does not require that probabilities sum up to 1:

**Definition 1** *The set of locally consistent pseudo marginals over $U$ is defined as:*

$$\mathcal{M}_L(U) = \{ \tilde{\boldsymbol{\mu}} \mid \sum_{x_i \in \bar{X}_i} \tilde{\mu}_{ij}(x_i, x_j) = \tilde{\mu}_j(x_j) \quad \forall (i, j) \in E, x_j \in \bar{X}_j \}.$$

*The partition function of $\tilde{\boldsymbol{\mu}}$, $Z(\tilde{\boldsymbol{\mu}})$, is given by $\sum_{x_i \in \bar{X}_i} \tilde{\mu}_i(x_i)$.*

The following theorem states that solving Eq. (5) is equivalent to solving an LP over $\mathcal{M}_L(U)$ with additional constraints.

**Theorem 4.2** *Let $U$ be a universe of assignments as defined in Eq. (6), $\boldsymbol{x} \in U$ and $\boldsymbol{\mu}$ a vector of tree-structured pairwise marginals, then the values of the following problems:*

$$\max_{p \in \mathcal{P}(\boldsymbol{\mu})} \sum_{\boldsymbol{u} \in U} p(\boldsymbol{u}), \ \max_{p \in \mathcal{P}(\boldsymbol{\mu})} \sum_{\boldsymbol{u} \in U \setminus (\boldsymbol{x}, \boldsymbol{y})} p(\boldsymbol{u}),$$

*are equal (respectively) to:*

$$\max_{\tilde{\boldsymbol{\mu}} \in \mathcal{M}_L(U), \tilde{\boldsymbol{\mu}} \leq \boldsymbol{\mu}} Z(\tilde{\boldsymbol{\mu}}), \ \max_{\substack{\tilde{\boldsymbol{\mu}} \in \mathcal{M}_L(U), \tilde{\boldsymbol{\mu}} \leq \boldsymbol{\mu} \\ I(\boldsymbol{x}, \boldsymbol{y}\,;\,\tilde{\boldsymbol{\mu}}) \leq 0}} Z(\tilde{\boldsymbol{\mu}}). \tag{7}$$

*These LPs involve a polynomial number of constraints and variables, thus can be solved efficiently.*

Proofs of this result can be obtained by exploiting properties of functions that decompose over trees. In the supplementary material, we provide a proof similar to that given in [30] to show equality of the marginal and local-marginal polytopes in tree models.

To conclude this section, we restate the main result: the robust conditional probability problem Eq. (2) can be solved in polynomial time by combining Theorems 4.1 and 4.2. As a by-product of this derivation we also presented efficient tools for bounding answers to a large class of probabilistic queries. While this is not the focus of the current paper, these tools may be a useful in probabilistic modelling, where we often combine estimates of low order marginals with assumptions on the data generating process. Bounds like the ones presented in this section give a quantitative estimate of the uncertainty that is induced by data and circumvented by our assumptions.

# 5 Closed Form Solutions and Combinatorial Algorithms

The results of the previous section imply that the minimum conditional can be found by solving a poly-sized LP. Although this results in polynomial runtime, it is interesting to improve as much as possible on the complexity of this calculation. One reason is that application of the bounds might require solving them repeatedly within some larger learning probelm. For instance, in classification tasks it may be necessary to solve Eq. (4) for each sample in the dataset. An even more demanding procedure will come up in our experimental evaluation, where we learn features that result in high confidence under our bounds. There, we need to solve Eq. (4) over mini-batches of training data only to calculate a gradient at each training iteration. Since using an LP solver in these scenarios is impractical, we next derive more efficient solutions for some special cases of Eq. (4).

## 5.1 Closed Form for Multiclass Problems

The multiclass setting is a special case of Eq. (4) when $y$ is a single label variable (e.g., a digit label in MNIST with values $y \in \{0, \ldots, 9\}$). The solution of course depends on the type of marginals provided in $\mathcal{P}(\boldsymbol{\mu})$. Here we will assume that we have access to joint marginals of the label $y$ and pairs of feature $x_i, x_j$ corresponding to edges $ij \in E$ of a graph $G$. We note that we can obtain similar results for the cases where some additional "unlabeled" statistics $\mu_{ij}(x_i, x_j)$ are known.

It turns out that in both cases Eq. (5) has a simple solution. Here we write it for the case without unlabeled statistics. The following lemma is based on a result that states $\max_{p \in \mathcal{P}(\boldsymbol{\mu})} p(\boldsymbol{x}) = \min_{ij} \mu_{ij}(x_i, x_j)$, which we prove in the supplementary material.

**Lemma 5.1** *Let $\boldsymbol{x} \in \mathcal{X}$ and $\boldsymbol{\mu}$ a vector of tree-structured pairwise marginals, then*

$$\min_{p \in \mathcal{P}(\boldsymbol{\mu})} p(y \mid \boldsymbol{x}) = \frac{I(\boldsymbol{x}, y\,;\,\boldsymbol{\mu})}{I(\boldsymbol{x}, y\,;\,\boldsymbol{\mu}) + \sum_{\bar{y} \neq y} \min_{ij} \mu_{ij}(x_i, x_j, \bar{y})}. \tag{8}$$

## 5.2 Combinatorial Algorithms and Connection to Maximum Flow Problems

In some cases, fast algorithms for the optimization problem in Eq. (5) can be derived by exploiting a tight connection of our problems to the Max-Flow problem. The problems are also closely related

to the weighted Set-Cover problem. To observe the connection to the latter, consider an instance of Set-Cover defined as follows. The universe is all assignments $\boldsymbol{x}$. Sets are defined for each $i, j, x_i, x_j$ and are denoted by $S_{ij,x_i,x_j}$. The set $S_{ij,x_i,x_j}$ contains all assignments $\bar{x}$ whose values at $i, j$ are $x_i, x_j$. Moreoever, the set $S_{ij,x_i,x_j}$ has weight $w(S_{ij,x_i,x_j}) = \mu_{ij}(x_i, x_j)$. Note that the number of items in each set is exponential, but the number of sets is polynomial. Now consider using these sets to cover some set of assignments $U$ with the minimum possible weight. It turns out that under the tree structure assumption, this problem is closely related to the problem of maximizing probabilities.

**Lemma 5.2** *Let $U$ be a set of assignments and $\boldsymbol{\mu}$ a vector of tree-structured marginals. Then:*

$$\max_{p \in \mathcal{P}(\boldsymbol{\mu})} \sum_{\boldsymbol{u} \in U} p(\boldsymbol{u}), \tag{9}$$

*has the same value as the standard LP relaxation [28] of the Set-Cover problem above.*

The connection to Set-Cover may not give a path to efficient algorithms, but it does illuminate some of the results presented earlier. It is simple to verify that $\min_{ij} \mu_{ij}(x_i, x_j, \bar{y})$ is a weight of a cover of $\boldsymbol{x}, \bar{y}$, while Eq. (3) equals one minus the weight of a set that covers all assignments but $\boldsymbol{x}, \boldsymbol{y}$. A connection that we may exploit to obtain more efficient algorithms is to Max-Flow. When the graph defined by $E$ is a chain, we show in the supplementary material that the value of Eq. (9) can be found by solving a flow problem on a simple network. We note that using the same construction, Eq. (5) turns out to be Max Flow under a budget constraint [1]. This may prove very beneficial for our goals, as it allows for efficient calculation of the robust conditionals we are interested in. Our conjecture is that this connection goes beyond chain graphs, but leave this for exploration in future work. The proofs for results in this section may also be found in the supplementary material.

# 6  Experiments

To evaluate the utility of our bounds, we consider their use in settings of semi-supervised deep learning and structured prediction. For the bounds to be useful, the marginal distributions need to be sufficiently informative. In some datasets, the raw features already provide such information, as we show in Section 6.3. In other cases, such as images, a single raw feature (i.e., a pixel) does not provide sufficient information about the label. These cases are addressed in Section 6.1 where we show how to learn new features which *do* result in meaningful bounds. Using deep networks to learn these features turns out to be an effective method for semi-supervised settings, reaching results close to those demonstrated by Variational Autoencoders [11]. It would be interesting to use such feature learning methods for structured prediction too; however this requires incorporation of the max-flow algorithm into the optimization loop, and we defer this to future work.

## 6.1  Deep Semi-Supervised Learning

A well known approach to semi-supervised learning is to optimize an empirical loss, while adding another term that measures prediction confidence on unlabeled data [9, 10]. Let us describe one such method and how to adapt it to use our bounds.

**Entropy Regularizer:** Consider training a deep neural network where the last layer has $n$ neurons $z_1, \dots, z_n$ connected to a softmax layer of size $|Y|$ (i.e. the number of labels), and the loss we use is a cross entropy loss. Denote the weights of the softmax layer by $W \in \mathbb{R}^{n \times |Y|}$. Given an input $\boldsymbol{x}$, define the softmax distribution at the output of the network as:

$$\tilde{p}_y = \text{softmax}_y(\langle W_y, z \rangle), \tag{10}$$

where $W_y$ is the $y$'th row of $W$. The min-entropy regularizer [9] adds an entropy term $\beta H(\tilde{p}_y)$ to the loss, for each unlabeled $\boldsymbol{x}$ in the training set.

**Plugging in Robust Conditional Probabilities:** We suggest a simple adaptation of this method that uses our bounds. Let us remove the softmax layer and set the activations of the neurons $z_1, \dots, z_n$ to a sigmoid activation. Let $Z_1, \dots, Z_n$ denote random variables that take on the values of the output neurons, these variables will be used as features in our bounds (in previous sections we refer to features as $X_i$. Here we switch to $Z_i$ since $X_i$ are understood as the raw features of the problem. e.g., the pixel values in the image). Since our bounds apply to discrete variables, while $z_1, \dots, z_n$ are real values, we use a smoothed version of our bounds.

**Loss Function and Smoothed Bounds:** A smoothed version of the marginals $\boldsymbol{\mu}$ is calculated by considering $Z_i$ as an indicator variable (e.g., the probability $p(Z_i = 1)$ would just be the average of the $Z_i$ values). Then the smoothed marginal $\bar{\mu}(z_i = 1, y)$ is the average of $z_i$ values over all training data labeled with $y$. In our experiments we used all the labeled data to estimate $\bar{\mu}$ at each iteration. The smoothed version of $I(\boldsymbol{z}, y; \boldsymbol{\mu})$, which we shall call $\bar{I}(\boldsymbol{z}, y; \boldsymbol{\mu})$, is then calculated with Eq. (3) when switching $\boldsymbol{\mu}$ with $\bar{\boldsymbol{\mu}}$ and the ReLU operator with a softplus.

To define a loss function we take a distribution over all labels:

$$\tilde{p}_y = \text{softmax}_y \left( \frac{\bar{I}(\boldsymbol{z}, y \,;\, \bar{\boldsymbol{\mu}})}{\bar{I}(\boldsymbol{z}, y \,;\, \bar{\boldsymbol{\mu}}) + \sum_{\bar{y} \neq y} \min_{ij} \bar{\mu}_{ij}(z_i, z_j, \bar{y})} \right) , \tag{11}$$

This is very similar to the standard distribution taken in a neural net, but it uses our bounds to make a more robust estimate of the conditionals. Then we use the exact same loss as the entropy regularizer, a cross entropy loss for labeled data with an added entropy term for unlabeled instances.

### 6.1.1 Algorithm Settings and Baselines

We implemented the min-entropy regularizer and our proposed method using a multilayer perceptron (MLP) with fully connected layers and a ReLU activation at each layer (except a sigmoid at the last layer for our method). In our experiments we used hidden layers of sizes $1000, 500, 50$ (so we learn $50$ features $Z_1, \ldots, Z_{50}$). We also add $\ell_2$ regularization on the weights of the soft-max layer for the entropy regularizer, since otherwise entropy can always be driven to zero in the separable case. We also experimented with adding a hinge loss as a regularizer (as in Transductive SVM [10]), but omit it from the comparison because it did not yield significant improvement over the entropy regularizer.

We also compare our results with those obtained by Variational Autoencoders and Ladder Networks. Although we do not expect to get accuracies as high as these methods, getting comparable numbers with a simple regularizer (compared to the elaborate techniques used in these works) like the one we suggest, shows that the use of our bounds results in a very powerful method.

### 6.2 MNIST Dataset

We trained the above models on the MNIST dataset, using $100$ and $1000$ labeled samples (see [11] for a similar setup). We set the two regularization parameters required for the entropy regularizer and the one required for our minimum probability regularizer with five fold cross validation. We used $10\%$ of the training data as a validation set and compared error rates on the $10^4$ samples of the test set. Results are shown in Figure 1. They show that on the 1000 sample case we are slightly outperformed by VAE and for 100 samples we lose by $1\%$. Ladder networks outperform other baselines.

| N | Ladder [21] | VAE [11] | Robust Probs | Entropy | MLP+Noise |
|---|---|---|---|---|---|
| 100 | 1.06($\pm$0.37) | 3.33($\pm$0.14) | 4.44($\pm$0.22) | 18.93($\pm$0.54) | 21.74($\pm$1.77) |
| 1000 | 0.84($\pm$0.08) | 2.40($\pm$0.02) | 2.48($\pm$0.03) | 3.15($\pm$0.03) | 5.70($\pm$0.20) |

Figure 1: Error rates of several semi-supervised learning methods on the MNIST dataset with few training samples.

**Accuracy vs. Coverage Curves:** In self-training and co-training methods, a classifier adds its most confident predictions to the training set and then repeats training. A crucial factor in the success of such methods is the error in the predictions we add to the training pool. Classifiers that use confidence over unlabelled data as a regularizer are natural choices for base classifiers in such a setting. Therefore an interesting comparison to make is the accuracy we would get over the unlabeled data, had the classifier needed to choose its $k$ most confident predictions.

We plot this curve as a function of $k$ for the entropy regularizer and our min-probabilities regularizer. Samples in the unlabelled training data are sorted in descending order according to confidence. Confidence for a sample in entropy regularized MLP is calculated based on the value of the logit that the predicted label received in the output layer. For the robust probabilities classifier, the confidence of a sample is the minimum conditional probability the predicted label received. As can be observed in Figure 6.2, our classifier ranks its predictions better than the entropy based method. We attribute this to our classifier being trained to give robust bounds under minimal assumptions.

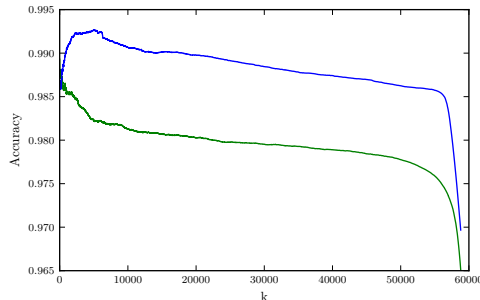

Figure 2: Accuracy for $k$ most confident samples in unlabelled data. Blue curve shows results for the Robust Probabilities Classifier, green for the Entropy Regularizer. Confidence is measured by conditional probabilities and logits accordingly.

### 6.3 Multilabel Structured Prediction

As mentioned earlier, in the structured prediction setting it is more difficult to learn features that yield high certainty. We therefore provide a demonstration of our method on a dataset where the raw features are relatively informative. The Genbase dataset taken from [26], is a protein classification multilabel dataset. It has 662 instances, divided into a training set of 463 samples and a test set of 199, each sample has 1185 binary features and 27 binary labels. We ran a structured-SVM algorithm, taken from [19] to obtain a classifier that outputs a labelling $\hat{y}$ for each $x$ in the dataset (the error of the resulting classifier was $2\%$). We then used our probabilistic bounds to rank the classifier's predictions by their robust conditional probabilities. The bounds were calculated based on the set of marginals $\mu_{ij}(x_i, y_j)$, estimated from the data for each pair of a feature and a label $X_i, Y_j$. The graph corresponding to these marginals is not a tree and we handled it as discussed in Section 7. The value of our bounds was above $0.99$ for $85\%$ of the samples, indicating high certainty that the classifier is correct. Indeed only $0.59\%$ of these $85\%$ were actually errors. The remaining errors made by the classifier were assigned a robust probability of $0$ by our bounds, indicating low level of certainty.

## 7 Discussion

We presented a method for bounding conditional probabilities of a distribution based only on knowledge of its low order marginals. Our results can be viewed as a new type of moment problem, bounding a key component of machine learning systems, namely the conditional distribution. As we show, calculating these bounds raises many challenging optimization questions, which surprisingly result in closed form expressions in some cases.

While the results were limited to the tree structured case, some of the methods have natural extensions to the cyclic case that still result in robust estimations. For instance, the local marginal polytope in Eq. (7) can be taken over a cyclic structure and still give a lower bound on maximum probabilities. Also in the presence of the cycles, it is possible to find the spanning tree that induces the best bound on Eq. (3) using a maximum spanning tree algorithm. Plugging these solutions into Eq. (4) results in a tighter approximation which we used in our experiments.

Our method can be extended in many interesting directions. Here we addressed the case of discrete random variables, although we also showed in our experiments how these can be dealt with in the context of continuous features. It will be interesting to calculate bounds on conditional probabilities given expected values of continuous random variables. In this case, sums-of-squares characterizations play a key role [15, 20, 3], and their extension to the conditional case is an exciting challenge. It will also be interesting to study how these bounds can be used in the context of unsupervised learning. One natural approach here would be to learn constraint functions such that the lower bound is maximized.

Finally, we plan to study the implications of our approach to diverse learning settings, from self-training to active learning and safe reinforcement learning.

**Acknowledgments:** This work was supported by the ISF Centers of Excellence grant 2180/15, and by the Intel Collaborative Research Institute for Computational Intelligence (ICRI-CI).

## Footnotes

[1]In the sense of minimizing prediction error.

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
