[Supplementary Material]

# Supplementary Material for Robust Conditional Probabilities

**Yoav Wald**
School of Computer Science and Engineering
Hebrew University
yoav.wald@mail.huji.ac.il

**Amir Globerson**
The Balvatnik School of Computer Science
Tel-Aviv University
gamir@mail.tau.ac.il

This document provides detailed proofs of theoretical results in the paper.

We first recall a property of functions that decompose over a tree structure. Assume we have a directed tree $G$ with $n$ nodes. Denote by $r$ its root, and by $pa(i)$ the parent of node $i$. Note that any undirected tree can be turned into a directed one by directing it away from an arbitrarily selected root. Now consider a function $\lambda(x_1, \ldots, x_n)$ over $n$ discrete variables. We will abbreviate $x_1, \ldots, x_n$ by $\boldsymbol{x}$ wherever clear from context. Assume that $\lambda(\boldsymbol{x})$ is defined as follows:

$$\lambda(\boldsymbol{x}) = \lambda_r(x_r) + \sum_{i \neq r} \lambda_{i,pa(i)}(x_i, x_{pa(i)}) + \lambda_i(x_i).$$

where $\lambda_r, \lambda_i$ and $\lambda_{i,j}$ are given singleton and pairwise functions. Then $\lambda(\boldsymbol{x})$ can be reparameterized using min "marginals", as defined below (See [2, 7] for proof of this result for max marginals and generalizations that include min operators):

$$\lambda(\boldsymbol{x}) = \bar{\lambda}_r(x_r) + \sum_{i \neq r} \bar{\lambda}_{i,pa(i)}(x_i, x_{pa(i)}) - \bar{\lambda}_{pa(i)}(x_{pa(i)}) \tag{0.1}$$

$$\bar{\lambda}_i(x_i) = \min_{\boldsymbol{z}:z_i=x_i} \lambda(\boldsymbol{z}), \ \bar{\lambda}_{ij}(x_i, x_j) = \min_{\boldsymbol{z}:z_i,z_j=x_i,x_j} \lambda(\boldsymbol{z})$$

Such $\lambda$ functions will arise, whenever we take the dual of a problem whose variables are a probability distribution constrained to satisfy some marginal distributions. Specifically, the multipliers $\lambda_i(x_i), \lambda_{ij}(x_i, x_j)$ will be those that correspond respectively to the primal constraints:

$$\sum_{\boldsymbol{z}:z_i=x_i} p(\boldsymbol{z}) = \mu_i(x_i), \quad \sum_{\boldsymbol{z}:z_i,z_j=x_i,x_j} p(\boldsymbol{z}) = \mu_{ij}(x_i, x_j).$$

## 1 Proof of Lem. 5.1

Let us begin with the proof of Lem. 5.1, in which we derive the form of solutions used in our experiments.

*Proof.* We start by writing the problem down in the following manner:

$$\min_{p \in \mathcal{P}(\boldsymbol{\mu})} \frac{p(\boldsymbol{x}, y)}{p(\boldsymbol{x}, y) + \sum_{\hat{y} \neq y} p(\boldsymbol{x}, \hat{y})}.$$

It is obvious that in order to minimize the objective, the higher $p(\boldsymbol{x}, \hat{y})$ is for $\hat{y} \neq y$ and the lower $p(\boldsymbol{x}, y)$, the lower objective we get. We now notice that each of the assignments can be maximized or minimized independently, because they appear in totally distinct constraints in $\mathcal{P}(\boldsymbol{\mu})$. This is true because all constraints in $\mathcal{P}(\boldsymbol{\mu})$ are of the form:

$$\sum_{\boldsymbol{z}:z_i,z_j=x_i,x_j} p(\boldsymbol{z}, \bar{y}) = \mu_{ij}(x_i, x_j, \bar{y}).$$

Hence, for any pair $y_1 \neq y_2$, non of the variables in $\{p(\boldsymbol{x}_1, y_1) \mid \boldsymbol{x}_1 \in \mathcal{X}\}$ appear in the same constraint with a variable in $\{p(\boldsymbol{x}_2, y_2) \mid \boldsymbol{x}_2 \in \mathcal{X}\}$, so all variables $p(\boldsymbol{x}, \hat{y}), p(\boldsymbol{x}, y)$ can be maximized or minimized separately. We already know from [5] that

$$\min_{p \in \mathcal{P}(\boldsymbol{\mu})} p(\boldsymbol{x}, y) = I(\boldsymbol{x}, y \,;\, \boldsymbol{\mu}).$$

It is left to show that

$$\max_{p \in \mathcal{P}(\boldsymbol{\mu})} p(\boldsymbol{x}, \bar{y}) = \min_{ij} \mu_{ij}(x_i, x_j, \bar{y}),$$

then the result of the lemma follows immediately. To prove the above equality we take the dual LP of the left hand side:

$$\min \ \boldsymbol{\lambda} \cdot \boldsymbol{\mu} \tag{1.1}$$
$$\text{s.t. } \lambda(\boldsymbol{x}, y) \geq 1$$
$$\lambda(\boldsymbol{z}, \bar{y}) \geq 0 \quad \forall \boldsymbol{z} \neq \boldsymbol{x} \vee \bar{y} \neq y.$$

Here $\lambda(\cdot)$ are the dual variables, which we can think of as a function that decomposes over a directed tree:

$$\lambda(\boldsymbol{x}, y) = \lambda_r(x_r, y) + \sum_{i \neq r} \lambda_{i,pa(i)}(x_i, x_{pa(i)}, y) + \lambda_i(x_i, y).$$

The inner product $\boldsymbol{\lambda} \cdot \boldsymbol{\mu}$ is given by:

$$\sum_{i, z_i} \lambda_i(z_i) \mu_i(z_i) + \sum_{ij \in E, z_i, z_j} \lambda_{ij}(z_i, z_j) \mu_{ij}(z_i, z_j). \tag{1.2}$$

Let us take the min-reparameterization of this function and then take its expectation over a distribution $p \in \mathcal{P}(\boldsymbol{\mu})$. The following inequality holds for any feasible $\lambda$:

$$\mathbb{E}_p\left[\lambda(\boldsymbol{x}, y)\right] = \sum_{z_r} \mu_r(z_r) \bar{\lambda}_r(z_r, y) + \sum_{\substack{i \neq r \\ z_i, z_{pa(i)}}} \mu_{i,pa(i)}(z_i, z_{pa(i)}) (\bar{\lambda}(z_i, z_{pa(i)}, y) - \bar{\lambda}_{pa(i)}(z_{pa(i)}, y))$$

$$\geq \mu_r(x_r) \bar{\lambda}_r(x_r, y) + \sum_{i \neq r} \mu_{i,pa(i)}(x_i, x_{pa(i)}) (\bar{\lambda}(x_i, x_{pa(i)}, y) - \bar{\lambda}_{pa(i)}(x_{pa(i)}, y)).$$

The inequality is true because any feasible $\lambda$ is non-negative, hence $\bar{\lambda}_r(z_r) \geq 0$ and because min-marginals over a pair of variables are always larger than those over one of them. We will conclude the proof by observing that:

- The right hand side of the inequality is a combination of the $\mu$s that are consistent with $\boldsymbol{x}, y$ and the coefficients of this combination sum up to:

$$\bar{\lambda}_r(x_r, y) + \sum_{i \neq r} \bar{\lambda}(x_i, x_{pa(i)}, y) - \bar{\lambda}_{pa(i)}(x_{pa(i)}, y) = \lambda(\boldsymbol{x}, y) \geq 1.$$

  The equality holds due to the reparametrization property in Eq. (0.1) and $\lambda$'s feasibility. Since the sum is higher than 1, the right hand side is also larger than any convex combination of the $\mu$s, which in turn is larger than the smallest element in the combination. We arrive at the conclusion that:
$$\mathbb{E}_p\left[\lambda(\boldsymbol{x}, y)\right] \geq \min_{ij} \mu_{ij}(x_i, x_j, y).$$

- It also holds that $\boldsymbol{\lambda} \cdot \boldsymbol{\mu} = \mathbb{E}_p\left[\lambda(x)\right]$, hence the objective of any feasible solution is larger than $\min_{ij} \mu_{ij}(x_i, x_j, y)$. On the other hand, setting $\lambda_{ij}(x_i, x_j, y) = 1$ for a minimizing pair $i, j$ and all other variables to 0 results in a feasible solution with exactly this objective. It follows that this must be the optimal value of the problem.

$\square$

## 2 Notations for Remainder of the Proofs

To allow for a more convenient notation, from now on we treat labels as hidden variables. That is, instead of $n$ features and $r$ labels, we assume there are just $n$ variables $X_1, \ldots, X_n$. The first $m$ are hidden (these will play the role of a label) and the last $n - m$ are observed, where $m$ may be between 0 and $n - 1$. For an assignment $x$, we refer to the hidden part as $x_h$ and the observed as $x_o$. The split into hidden and observed variables will mainly serve us in the proof of Thm. 4.1, in other proofs it is just more convenient to not split expressions to $x, y$.

We also denote the subvector of $\mu$ over hidden variables and edges between them as $\mu_h$. That is, considering the items of $\mu$ are expressions $\mu_i(z_i), \mu_{ij}(z_i, z_j)$, $\mu_h$ is the subvector containing items where $i \in h, i, j \in h$ respectively. Define a similar vector $\mu_o$ for observed variables and edges between them. The vectors $\mathbb{I}_x, \mathbb{I}_{h,x}$ are defined to have the same indices as $\mu, \mu_h$ respectively, their value is 1 in indices consistent with $x$ (i.e. $z_i, z_j = x_i, x_j$ or $z_i = x_i$ for entries that contain $\mu_{ij}(z_i, z_j), \mu_i(z_i)$ respectively) and 0 otherwise. We will use the shorthand $\mathbf{I}_x$ for the vector $I(x; \mu)\mathbb{I}_x$.

Some notations related to graphical properties of hidden and observed nodes will be required. The number of connected components in the subgraph of hidden variables and edges between them is $|P_h|$, similarly for observed variables we will use $|P_o|$. The set of edges $ij$ between hidden nodes (i.e. $i, j \in h$) is $E_h$, between a hidden and observed node (i.e. $i \in o, j \in h$ w.l.o.g) is $E_{oh}$ and between observed nodes (i.e. $i, j \in o$) is $E_o$. The degree of node $i$ is $d_i$ and the number of its hidden neighbors is $d_i^h$.

Finally, we define variations on the objects related to graphical models that we use in the paper. The functional $\tilde{I}(\cdot; \mu)$ is the same functional defined in Eq. (3) of the paper, only without the ReLU operator:

$$\tilde{I}(x; \mu) = \sum_i (1 - d_i)\mu_i(x_i) + \sum_{ij \in E} \mu_{ij}(x_i, x_j).$$

We will also use two variants on the local marginal polytope [7]:

$$\mathcal{M}_L = \left\{ \tilde{\mu} \mid \begin{array}{ll} \sum_{x_j} \tilde{\mu}_{ij}(x_i, x_j) = \tilde{\mu}_i(x_i) & \forall ij \in E, x_i \\ \sum_{x_i} \tilde{\mu}_{ij}(x_i, x_j) = \tilde{\mu}_j(x_j) & \forall ij \in E, x_j \end{array}, \begin{array}{ll} \sum_{x_i} \tilde{\mu}_i(x_i) = 1 & \forall i \\ \sum_{x_i, x_j} \tilde{\mu}_i(x_i, x_j) = 1 & \forall i, j \in E \end{array} \right\}.$$

One variant we use is $\mathcal{M}_L(U)$ that was defined in the paper. The other is $\mathcal{M}_L^h$, where items contain marginals only on hidden variables and edges between them:

$$\mathcal{M}_L^h = \left\{ \tilde{\mu} \mid \begin{array}{ll} \sum_{x_i \in \bar{X}_i} \tilde{\mu}_{ij}(x_i, x_j) = \tilde{\mu}_j(x_j) & \forall (i, j) \in E_h \\ \sum_{x_j \in \bar{X}_j} \tilde{\mu}_{ij}(x_i, x_j) = \tilde{\mu}_i(x_i) & \forall (i, j) \in E_h \end{array} \right\}.$$

## 3 Proof of Lem. 5.2

We start by proving the connection to Set-Cover and then move on to Max-Flow.

### 3.1 Connection to Set-Cover

*Proof.* Considrt Eq. (9) of the paper and let us write down its dual:

$$\min \; \boldsymbol{\lambda} \cdot \boldsymbol{\mu} \tag{3.1}$$

$$\text{s.t. } \lambda_r(x_r) + \sum_{i \neq r} \lambda_{i, pa(i)}(x_i, x_{pa(i)}) + \lambda_i(x_i) \geq 0 \quad \forall x \notin U$$

$$\lambda_r(x_r) + \sum_{i \neq r} \lambda_{i, pa(i)}(x_i, x_{pa(i)}) + \lambda_i(x_i) \geq 1 \quad \forall x \in U,$$

This is already very similar to the LP Relaxation of Set-Cover, but with the significant difference that variables $\lambda$ are unrestricted, where in the Set-Cover LP they are non-negative. This is where the tree structure plays an important role. Consider the min-reparameterization of any feasible solution $\lambda(x)$:

$$\lambda(x) = \bar{\lambda}_r(x_r) + \sum_{i \neq r} \bar{\lambda}_{i, pa(i)}(x_i, x_{pa(i)}) - \bar{\lambda}_{pa(i)}(x_{pa(i)}).$$

Since $\boldsymbol{\lambda}$ is feasible and the constraints demand that $\lambda(\boldsymbol{x})$ is non negative for all $\boldsymbol{x}$, it is clear that $\bar{\lambda}_r(x_r) \geq 0$. Moreover, because $\bar{\lambda}$ is a min-reparameterization it is easy to see that $\bar{\lambda}_{i,pa(i)}(x_i, x_{pa(i)}) - \bar{\lambda}_{pa(i)}(x_{pa(i)}) \geq 0$. This is true because constraining a minimization on $x_i, x_{pa(i)}$ gives a higher result than constraining on $x_{pa(i)}$ alone.

Now let us look at the LP Relaxation of the aforementioned Set-Cover problem:

$$\min \; \boldsymbol{\delta} \cdot \boldsymbol{\mu} \tag{3.2}$$

$$\text{s.t. } \delta_r(x_r) + \sum_{i \neq r} \delta_{i,pa(i)}(x_i, x_{pa(i)}) + \delta_i(x_i) \geq 0 \quad \forall \boldsymbol{x} \notin U$$

$$\delta_r(x_r) + \sum_{i \neq r} \delta_{i,pa(i)}(x_i, x_{pa(i)}) + \delta_i(x_i) \geq 1 \quad \forall \boldsymbol{x} \in U,$$

$$\boldsymbol{\delta} \geq 0$$

Obviously, if $\boldsymbol{\delta}$ is feasible for Eq. (3.2), setting $\boldsymbol{\lambda} = \boldsymbol{\delta}$ gives a feasible solution to Eq. (3.1) with the same objective as $\boldsymbol{\delta}$'s in Eq. (3.2). That is, this problem is more constrained than Eq. (3.1). Yet given a feasible solution to Eq. (3.1), we can use the min-reparameterization and obtain a feasible solution to the above problem with the same objective $\boldsymbol{\lambda} \cdot \boldsymbol{\mu}$:

$$\delta_i(x_i) = \begin{cases} \bar{\lambda}_r(x_r) & i = r \\ 0 & i \neq r \end{cases}, \quad \delta_{i,pa(i)}(x_i, x_{pa(i)}) = \bar{\lambda}_{i,pa(i)}(x_i, x_{pa(i)}) - \bar{\lambda}_{pa(i)}(x_{pa(i)}).$$

It is easy to see that because of the min-reparameteriztion property, $\delta(\boldsymbol{x}) = \lambda(\boldsymbol{x})$ for all $\boldsymbol{x}$ and $\boldsymbol{\delta} \geq 0$. This means that $\boldsymbol{\delta}$ is feasible and that the objectives are equal. To verify the latter, consider a distribution $p \in \mathcal{P}(\boldsymbol{\mu})$. Taking the expectations of $\boldsymbol{\delta}, \boldsymbol{\mu}$ with respect to $p$ shows the equality in objectives:

$$\boldsymbol{\lambda} \cdot \boldsymbol{\mu} = \mathbb{E}_p[\lambda(\boldsymbol{x})] = \mathbb{E}_p[\delta(\boldsymbol{x})] = \boldsymbol{\delta} \cdot \boldsymbol{\mu}.$$

We conclude that while the set cover LP Relaxation is more constrained, all feasible solutions of Eq. (3.1) can be mapped to feasible solutions of this relaxation in a manner that preserves the objective. Hence the problems have the same value. $\qquad \square$

Let us emphasize the following two points:

- This part of the lemma did not exploit the specific choice of $U$ (being consisted of all assignments where variables take values in a certain set $\bar{X}_i$). That is, it holds for any choice of $U$, not only those of the form mentioned in Eq. (6).
- The constraints for $\boldsymbol{x} \notin U$ in Eq. (3.2) are redundant because $\boldsymbol{\delta} \geq 0$. Removing these constraints and moving back from Eq. (3.2) to its dual, expressed with variables $p$, we get another formulation of Eq. (9). We will use this in the next part of the proof and also later on, we thus state it as a corollary.

**Corollary 3.1.** *Let $U$ be a universe of assignments (not necessarily of the form in Eq. (6)) and $\boldsymbol{\mu}$ a tree-structured vector of marginals. The following LP has the same value as Eq. (9):*

$$\max_{p \geq 0} \sum_{\boldsymbol{u} \in U} p(\boldsymbol{u}) \tag{3.3}$$

$$\text{s.t.} \sum_{\substack{\boldsymbol{u} \in U \\ u_i, u_j = z_i, z_j}} p(\boldsymbol{u}) \leq \mu_{i,j}(z_i, z_j) \qquad \forall ij \in E, z_i, z_j$$

$$\sum_{\substack{\boldsymbol{u} \in U \\ u_i = z_i}} p(\boldsymbol{u}) \leq \mu_i(z_i) \qquad \forall i \in V, z_i$$

### 3.2 Equivalence to Max-Flow

As stated in Section 5.2 of the paper, when the underlying graph is a chain, Eq. (9) is a Max-Flow problem. The equivalence to Max-Flow is apparent when thinking of every assignment $\boldsymbol{x} \in U$ as a path in a flow network. Assume our statistics $\boldsymbol{\mu}$ are $\mu_{1,2}, \mu_{2,3}, \dots, \mu_{n-1,n}$, then define a flow

network with source and sink $s, t$ and a node $(i, x_i)$ for each variable $i$ and $x_i \in \bar{X}_i$ (i.e. one node for each variable-assignment pair). The edges of the network are $(i, x_i) \to (i+1, x_{i+1})$ for each $0 \leq i \leq n-1$ and $x_i, x_{i+1} \in \bar{X}_i \times \bar{X}_{i+1}$, they will have capacity $\mu_{i,i+1}(x_i, x_{i+1})$. Additionally we will have edges $s \to (1, x_1), (n, x_n) \to t$ for each $x_1$ and $x_n$ with unbounded capacity.

It is simple to see that there is a one-to-one correspondence between paths from $s$ to $t$ and assignments in $U$. This is where $U$'s special structure, stated in Eq. (6) of the paper comes into play. Also, the paths that go through each edge $(i, x_i) \to (i+1, x_{i+1})$ are exactly those of assignments $\boldsymbol{z}$ where $z_i, z_{i+1} = x_i, x_{i+1}$. According to flow decomposition [4], the LP in Eq. (3.3) solves the Max-Flow problem on this network (where the flow is expressed as the sum of flows in all $s - t$ paths in the network), with a single exception that it does not contain the constraints:

$$\sum_{\substack{\boldsymbol{u} \in U \\ u_i = z_i}} p(\boldsymbol{u}) \leq \mu_i(z_i) \quad \forall i \in V, z_i.$$

Thus to finish the proof we will get convinced that these added constraints are redundant. Consider a solution $p$ that only satisfies the constraints of pairwise marginals in Eq. (3.3), we will show it also satisfies the constraints above. Let $i \in [n]$ and $x_i \in \bar{X}_i$ and let $j$ be a neighbour of $i$ in the chain (the graph is connected, so there always is a neighbour), then:

$$\sum_{\substack{\boldsymbol{u} \in U \\ u_i = x_i}} p(\boldsymbol{u}) = \sum_{u_j \in \bar{X}_j} \sum_{\substack{\boldsymbol{u} \in U \\ u_i, u_j = x_i, x_j}} p(\boldsymbol{u}) \leq \sum_{u_j \in \bar{X}_j} \mu_{ij}(x_i, u_j) \leq \mu_i(x_i).$$

This shows the constraint is satisfied and concludes our proof.

The next proof, that of Thm. 4.2, is for results on maximizing probabilities. When the underlying graph is a chain, these results are similar to the equivalence to Max-Flow that we just proved. When the graph is not a chain, they will give an LP that does not directly correspond to a Max-Flow problem, but is still of polynomial size. That is, it can be solved efficiently with a standard LP solver, but not necessarily with a combinatorial algorithm. Our conjecture is that combinatorial algorithms can be derived for other cases, but we defer this to future work.

## 4 Proof of Thm. 4.2

The theorem reformulates the following problems:

$$\max_{p \in \mathcal{P}(\boldsymbol{\mu})} \sum_{\boldsymbol{u} \in U} p(\boldsymbol{u}), \ \max_{p \in \mathcal{P}(\boldsymbol{\mu})} \sum_{\boldsymbol{u} \in U \setminus \boldsymbol{x}} p(\boldsymbol{u}). \tag{4.1}$$

Our goal is to show that they have the same optimum as:

$$\max_{\tilde{\boldsymbol{\mu}} \in \mathcal{M}_L(U), \tilde{\boldsymbol{\mu}} \leq \boldsymbol{\mu}} Z(\tilde{\boldsymbol{\mu}}), \ \max_{\substack{\tilde{\boldsymbol{\mu}} \in \mathcal{M}_L(U), \tilde{\boldsymbol{\mu}} \leq \boldsymbol{\mu} \\ I(\boldsymbol{x}\,;\,\tilde{\boldsymbol{\mu}}) \leq 0}} Z(\tilde{\boldsymbol{\mu}}). \tag{4.2}$$

*Proof.* To show equality of the optimal values, let us offer a mapping between feasible solutions of the pairs of problems. From our previous results, both problems in Eq. (4.1) can be written in the form of Eq. (3.3) with $U$ and $U \setminus \boldsymbol{x}$ respectively. We will start by mapping feasible solutions of these problems to feasible solutions of Eq. (4.2).

Choose an arbitrary root for the tree, $r \in V$, and turn the undirected tree to a directed one rooted in $r$. Consider a feasible solution $p$ to the reformulated problem in Eq. (3.3) and define:

$$\tilde{\mu}_{i,pa(i)}(u_i, u_{pa(i)}) = \sum_{\boldsymbol{z} \in U: z_i, z_{pa(i)} = u_i, u_{pa(i)}} p(\boldsymbol{z}) \qquad \forall (u_i, u_{pa(i)}) \in \bar{X}_i \times \bar{X}_{pa(i)}$$

$$\tilde{\mu}_i(u_i) = \sum_{\boldsymbol{z} \in U: z_i = u_i} p(\boldsymbol{z}) \qquad \forall u_i \in \bar{X}_i$$

It is simple to prove that $\tilde{\boldsymbol{\mu}} \in \mathcal{M}_L(U)$, because for any pair $ij \in E$ it holds that:

$$\sum_{u_j \in \bar{X}_j} \tilde{\mu}_{i,j}(u_i, u_j) = \sum_{u_j \in \bar{X}_j} \sum_{\boldsymbol{z} \in U: z_i, z_j = u_i, u_j} p(\boldsymbol{z}) = \sum_{\boldsymbol{z} \in U: z_i = u_i} p(\boldsymbol{z}) = \tilde{\mu}_i(u_i).$$

And from $p$'s feasibility we also get $\tilde{\boldsymbol{\mu}} \leq \boldsymbol{\mu}$. This can be seen from inequalities of the following type:

$$\tilde{\mu}_{i,j}(u_i, u_j) = \sum_{\boldsymbol{z} \in U : z_i, z_j = u_i, u_j} p(\boldsymbol{z}) \leq \mu_{ij}(u_i, u_j).$$

We conclude that $\tilde{\boldsymbol{\mu}}$ is a feasible solution to Eq. (4.2) with objective:

$$Z(\tilde{\boldsymbol{\mu}}) = \sum_{u_r \in \bar{X}_r} \tilde{\mu}_r(z_r) = \sum_{u_r \in \bar{X}_r} \sum_{\boldsymbol{z} \in U : z_r = u_r} p(\boldsymbol{z}) = p(U).$$

This mapping only considered the first problem in Eq. (4.1). We can use the exact same construction when considering $U \setminus \boldsymbol{x}$ as follows. Feasible solutions to Eq. (3.3) are functions $p : U \setminus \boldsymbol{x} \to \mathbb{R}_+$, so extending $p$'s domain to $U$ by setting $p(\boldsymbol{x}) = 0$, the above equations remain unaltered. It is left to show that the resulting $\tilde{\boldsymbol{\mu}}$ satisfies $I(\boldsymbol{x}; \tilde{\boldsymbol{\mu}}) \leq 0$. If we examine the term $I(\boldsymbol{x}; \tilde{\boldsymbol{\mu}})$, when $d_i$ is the degree of node $i$ in the graph, we get that:

$$\sum_i (1 - d_i)\tilde{\mu}_i(x_i) + \sum_{ij} \tilde{\mu}_{ij}(x_i, x_j) = \sum_{\boldsymbol{u} \in U} \alpha_{\boldsymbol{u}} p(\boldsymbol{u}),$$

$$\alpha_{\boldsymbol{u}} \triangleq \sum_i \mathbb{I}_{u_i = x_i} - \sum_{ij} \mathbb{I}_{(u_i = x_i) \vee (u_j = x_j)}.$$

Simple counting arguments show that $\alpha_{\boldsymbol{x}} = 1$, while $\alpha_{\boldsymbol{u}} \leq 0$ for all $\boldsymbol{u} \neq \boldsymbol{x}$. Since we set $p(\boldsymbol{x}) = 0$, it follows that $\sum_{\boldsymbol{u} \in U} \alpha_{\boldsymbol{u}} p(\boldsymbol{u}) \leq 0$ and also $I(\boldsymbol{x}; \tilde{\boldsymbol{\mu}})$.

It is left to provide a mapping from solutions of Eq. (4.2) to solutions of Eq. (4.1). We will provide a proof for the case where

$$U = \left\{ \boldsymbol{u} \mid u_i \in \bar{X}_i \quad \forall i \in [n] \right\}.$$

More specifically, we will construct a function $p : U \to \mathbb{R}_+$ whose marginals are $\tilde{\boldsymbol{\mu}}$ and summing it over all of its domain gives $Z(\tilde{\boldsymbol{\mu}})$. The construction is the same one used when proving that the local marginal polytope is equal to the marginal polytope for tree graphs [7]. To complete the proof, we will also need to show a construction when $p$'s domain is $U \setminus \boldsymbol{x}$ (and $U$ defined the same as above). We refer the reader to [5] where this detailed construction can be found. There the sum of $p$ over its domain is 1, yet applying this construction to $\tilde{\boldsymbol{\mu}}$ gives a function that sums up to $Z(\tilde{\boldsymbol{\mu}})$.

The function $p$ we suggest for the problem over domain $U$ is:

$$p(\boldsymbol{u}) = \tilde{\mu}_r(u_r) \prod_{i \neq r} \frac{\tilde{\mu}_{i, pa(i)}(u_i, u_{pa(i)})}{\tilde{\mu}_{pa(i)}(u_i)}.$$

Assume $r$ is set arbitrarily and $1, \ldots, n$ is a topological ordering of the nodes. Notice that any choice of $r$ and an ordering yields the same function $p$. It is simple to see that the function marginalizes to $\tilde{\mu}$ if we let $ij \in E$, set $i$ as the root and eliminate all variables other than $i, j$. To show that $p$'s sum over its domain $U$ is exactly the partition function, eliminate all the variables to get:

$$\sum_{\boldsymbol{x} \in U} p(\boldsymbol{x}) =$$

$$\sum_{u_1 \in \bar{X}_1} \tilde{\mu}_1(u_1) \left( \sum_{u_2 \in \bar{X}_2} \frac{\tilde{\mu}_{2, pa(2)}(u_2, u_{pa(2)})}{\tilde{\mu}_{pa(2)}(u_2)} \cdots \left( \sum_{u_n \in \bar{X}_n} \frac{\tilde{\mu}_{n, pa(n)}(u_n, u_{pa(n)})}{\tilde{\mu}_{pa(n)}(u_{pa(n)})} \right) \right) = \sum_{u_1 \in \bar{X}_1} \tilde{\mu}_1(u_1).$$

Here we implicitly numbered the root node as 1. To conclude, we showed a mapping from $\tilde{\mu}$ to a function $p$ that is feasible for Eq. (4.1), completing the proof.

For the case $U \setminus \boldsymbol{x}$, as stated earlier, [5] offer a construction of a function that marginalizes to $\tilde{\mu}$ and achieves $p(\boldsymbol{x}) = I(\boldsymbol{x} ; \boldsymbol{\mu})$. Thus enforcing $I(\boldsymbol{x} ; \boldsymbol{\mu}) \leq 0$ ensures there is a mapping from $\tilde{\boldsymbol{\mu}}$ to a function $p$ with the same objective.

Notice the equality in the above equation holds because of $U$'s special structure that includes **all** the assignments that take values in sets $\bar{X}_i$. Different choices of $U$ do not necessarily yield this equation, thus the theorem does not hold for all choices of $U$. $\qquad \square$

# 5 Proof of Thm. 4.1

We recall the problem at hand of minimizing conditional probabilities:

$$\min_{p \in \mathcal{P}(\boldsymbol{\mu})} p(\boldsymbol{x}_h \mid \boldsymbol{x}_o),$$

where we assume w.l.o.g that $\boldsymbol{x}_h = x_1, \ldots, x_m$ are hidden variables, $\boldsymbol{x}_o = x_{m+1}, \ldots, x_n$ are observed, and $\boldsymbol{x}$ is the fixed assignment to both. Using the Charnes-Cooper variable transformation [1] between $p(\boldsymbol{z}_h, \boldsymbol{z}_o)$ and $\frac{p(\boldsymbol{z}_h, \boldsymbol{z}_o)}{p(\boldsymbol{x}_o)}$ for all $\boldsymbol{z}$, and taking the dual of the resulting LP, we arrive at the following problem:

$$\max \ \lambda_{\boldsymbol{x}} \tag{5.1}$$

$$\text{s.t. } \lambda_r(z_r) + \sum_{i \neq r} \lambda_{i,pa(i)}(z_i, z_{pa(i)}) + \lambda_i(z_i) \leq 0 \qquad \forall \boldsymbol{z} : \ \boldsymbol{z}_o \neq \boldsymbol{x}_o$$

$$\lambda_r(z_r) + \sum_{i \neq r} \lambda_{i,pa(i)}(z_i, z_{pa(i)}) + \lambda_i(z_i) \leq -\lambda_{\boldsymbol{x}} \qquad \forall \boldsymbol{z} : \ \boldsymbol{z}_o = \boldsymbol{x}_o, \boldsymbol{z}_h \neq \boldsymbol{x}_h,$$

$$\lambda_r(x_r) + \sum_{i \neq r} \lambda_{i,pa(i)}(x_i, x_{pa(i)}) + \lambda_i(x_i) \leq 1 - \lambda_{\boldsymbol{x}}$$

$$\boldsymbol{\lambda} \cdot \boldsymbol{\mu} \geq 0.$$

The transformation is correct under the assumption that $p(\boldsymbol{x}_o) > 0$, which is reasonable to assume when we observe $\boldsymbol{x}_o$ and try to infer $\boldsymbol{x}_h$.

The rest of the proof can now be decomposed into two main parts, one manipulates Eq. (5.1) and the other manipulates the second problem in Eq. (4.2):

**Lemma 5.1.** *Let $U$ be a set of the shape defined in Eq. (6) of the paper and $\boldsymbol{\mu}$ a vector of tree shaped marginals. If*

$$\max_{p \in \mathcal{P}(\boldsymbol{\mu})} \sum_{\boldsymbol{u} \in U} p(\boldsymbol{u}) > \max_{p \in \mathcal{P}(\boldsymbol{\mu})} \sum_{\boldsymbol{u} \in U \setminus \boldsymbol{x}} p(\boldsymbol{u}), \tag{5.2}$$

*then it holds that:*

$$\max_{\substack{\tilde{\boldsymbol{\mu}} \in \mathcal{M}_L(U), \tilde{\boldsymbol{\mu}} \leq \boldsymbol{\mu} \\ I(\boldsymbol{x}\,;\,\tilde{\boldsymbol{\mu}}) \leq 0}} Z(\tilde{\boldsymbol{\mu}}) = \max_{\substack{\tilde{\boldsymbol{\mu}} \in \mathcal{M}_L(U), \tilde{\boldsymbol{\mu}} \leq \boldsymbol{\mu} - \mathbf{I}_{\boldsymbol{x}} \\ I(\boldsymbol{x}\,;\,\tilde{\boldsymbol{\mu}}) = 0}} Z(\tilde{\boldsymbol{\mu}}).$$

**Lemma 5.2.** *Eq. (5.1) has the same optimal value as:*

$$\min \ \mu_{\boldsymbol{x}} \tag{5.3}$$

$$\text{s.t. } \tilde{\boldsymbol{\mu}} \in \mathcal{M}_L^h, 0 \leq \tilde{\boldsymbol{\mu}} \leq \tau_{\mu} \boldsymbol{\mu}_h - \mu_{\boldsymbol{x}} \mathbb{I}_{\boldsymbol{x}}$$

$$\boldsymbol{\mu}_o \tau_{\mu} \geq 1$$

$$\sum_{z_i} \tilde{\mu}_i(z_i) = \tilde{\tau} \quad \forall i \in h$$

$$\mu_{\boldsymbol{x}} + \tilde{\tau} = 1$$

$$I(\boldsymbol{x}_h; \tilde{\boldsymbol{\mu}}) + (1 - |P_h|)\tilde{\tau} \leq 0$$

$$\tau_{\mu} I(\boldsymbol{x}; \boldsymbol{\mu}) - \mu_{\boldsymbol{x}} - I(\boldsymbol{x}_h; \tilde{\boldsymbol{\mu}}) + (|P_h| - 1)\tilde{\tau} \leq 0.$$

The decision variables in in Eq. (5.3) are $\tilde{\boldsymbol{\mu}}, \tilde{\tau}, \tau_{\mu}, \mu_{\boldsymbol{x}}$, where $\tilde{\boldsymbol{\mu}}$ are pseudo-marginals on hidden variables and pairs of them that are connected by an edge. This form is very similar to that of problems in Eq. (4.2), and indeed their solutions are similar. Using Lem. 5.1, we will show that a simple modification to the solution of the second problem in Eq. (4.2) leads to a solution of Eq. (5.3). This modification is shown in the following two lemmas, that also conclude the proof of Thm. 4.1. For now we assume the correctness of Lem. 5.2 and Lem. 5.1, their proofs are deferred to the end of this document.

To fit our problem into the formulation of Lem. 5.1, define $U$ using $\bar{X}_i = \{x_i\}$ for all observed variables $i \in o$ and $\bar{X}_j$ unrestricted for all hidden variables $j \in h$. Under this definition we have:

$$\max_{p \in \mathcal{P}(\boldsymbol{\mu})} \sum_{\boldsymbol{u} \in U} p(\boldsymbol{u}) = \max_{p \in \mathcal{P}(\boldsymbol{\mu})} p(\boldsymbol{x}_o),$$

$$\max_{p \in \mathcal{P}(\boldsymbol{\mu})} \sum_{\boldsymbol{u} \in U \setminus \boldsymbol{x}} p(\boldsymbol{u}) = \max_{p \in \mathcal{P}(\boldsymbol{\mu})} \sum_{\boldsymbol{z}_h \neq \boldsymbol{x}_h} p(\boldsymbol{x}_o, \boldsymbol{z}_h).$$

We are now ready to use the above lemmas and conclude the proof.

**Lemma 5.3.** *If* $I(\boldsymbol{x} \,;\, \boldsymbol{\mu}) \leq 0$ *then*

$$\min_{p \in \mathcal{P}(\boldsymbol{\mu})} p(\boldsymbol{x}_h \mid \boldsymbol{x}_o) = 0,$$

*unless* $\max_{p \in \mathcal{P}(\boldsymbol{\mu})} \sum_{\boldsymbol{z}_h \neq \boldsymbol{x}_h} p(\boldsymbol{z}_h, \boldsymbol{x}_o) = 0$ *and then the value is* $1$.

*Proof.* We assume that $p(\boldsymbol{x}_o)$ is constrained to be larger than $0$, otherwise the robust conditional probability problem is ill-defined. So it is trivial that if

$$\max_{p \in \mathcal{P}(\boldsymbol{\mu})} \sum_{\boldsymbol{z}_h \neq \boldsymbol{x}_h} p(\boldsymbol{x}_o, \boldsymbol{z}_h) = 0,$$

then $p(\boldsymbol{x}) = p(\boldsymbol{x}_o)$ and the conditional is $1$.

Now assume towards contradiction that $\min_{p \in \mathcal{P}(\boldsymbol{\mu})} p(\boldsymbol{x}_h \mid \boldsymbol{x}_o) > 0$, clearly we must have:

$$\max_{p \in \mathcal{P}(\boldsymbol{\mu})} \sum_{\boldsymbol{u} \in U} p(\boldsymbol{u}) > \max_{p \in \mathcal{P}(\boldsymbol{\mu})} \sum_{\boldsymbol{u} \in U \setminus \boldsymbol{x}} p(\boldsymbol{u}),$$

because otherwise equality must hold, so a maximizing distribution of the right hand side will have to achieve a conditional probability of $0$. Then the conditions of Lem. 5.1 hold and we have:

$$\max_{p \in \mathcal{P}(\boldsymbol{\mu})} \sum_{\boldsymbol{z}_h \neq \boldsymbol{x}_h} p(\boldsymbol{x}_o, \boldsymbol{z}_h) = \max_{\substack{\tilde{\boldsymbol{\mu}} \in \mathcal{M}_L(U), \tilde{\boldsymbol{\mu}} \leq \boldsymbol{\mu} \\ I(\boldsymbol{x} \,;\, \tilde{\boldsymbol{\mu}}) = 0}} Z(\tilde{\boldsymbol{\mu}}).$$

Denote the value of the above problems as $\tilde{\tau}_1 > 0$, let $\tilde{\boldsymbol{\mu}}_1$ be an optimal solution to the problem on the right hand side and $\tilde{\boldsymbol{\mu}}_{1,h}$ its sub-vector that corresponds to hidden variables and edges between them. Consider taking $\tilde{\boldsymbol{\mu}} = \frac{\tilde{\boldsymbol{\mu}}_{1,h}}{\tilde{\tau}_1}, \tilde{\tau} = 1, \mu_{\boldsymbol{x}} = 0$, we will show there exists a value of $\tau_{\mu}$ such that $\tilde{\boldsymbol{\mu}}, \tilde{\tau}, \mu_{\boldsymbol{x}}, \tau_{\mu}$ is a feasible solution to Eq. (5.3). The value of this solution is $\mu_{\boldsymbol{x}} = 0$, which contradicts the assumption that the minimum is strictly positive and concludes the proof.

To see such a value of $\tau_{\mu}$ exists, note the following three points:

- $\tilde{\boldsymbol{\mu}}_1 \in \mathcal{M}_L(U), \tilde{\boldsymbol{\mu}}_1 \leq \boldsymbol{\mu}$ and normalizes to $\tilde{\tau}_1$. So it also holds that $\tilde{\boldsymbol{\mu}} \in \mathcal{M}_L, \tilde{\boldsymbol{\mu}} \leq \tilde{\tau}_1^{-1} \boldsymbol{\mu}_h$, hence the first constraint of Eq. (5.3) is satisfied for any $\tau_{\mu} \geq \tilde{\tau}_1^{-1}$. Also from these results it is straightforward to see that the third and fourth constraints are satisfied.

- Because we enforced $p(\boldsymbol{x}_o) > 0$, it holds that $\boldsymbol{\mu}_o > 0$. Thus the second constraint of Eq. (5.3) can also be satisfied if we take a large enough value for $\tau_{\mu}$ (i.e. larger than one over the minimal item in $\boldsymbol{\mu}_o$).

- Finally, we will show that

$$I(\boldsymbol{x}_h; \tilde{\boldsymbol{\mu}}) + (1 - |P_h|)\tilde{\tau} = 0. \tag{5.4}$$

  This means the fifth constraint is satisfied and more importantly, because $I(\boldsymbol{x}; \boldsymbol{\mu}) \leq 0$, the last constraint is satisfied for any positive value of $\tau_{\mu}$.

  To show that Eq. (5.4) holds, notice that:

$$
\begin{aligned}
I(\boldsymbol{x}; \tilde{\boldsymbol{\mu}}_1) &= 0, & \\
\tilde{\mu}_{1,i}(x_i) &= \tilde{\tau}_1 & \forall i \in o, \\
\tilde{\mu}_{1,ij}(x_i, x_j) &= \tilde{\tau}_1 & \forall (i,j) \in E_o, \\
\tilde{\mu}_{1,ij}(x_i, z_j) &= \tilde{\mu}_{1,j}(z_j) & \forall (i,j) \in E_{oh}, z_j.
\end{aligned}
$$

This first equality holds because it is a constraint in the problem that $\tilde{\mu}_1$ solves, the others because observed variables have only one possible value in $U$ and $\tilde{\mu}_1 \in \mathcal{M}_L(U)$. Let us write down $I(\boldsymbol{x}; \tilde{\boldsymbol{\mu}}_1)$ and decompose the sums in its expression into smaller ones over observed and hidden variables, and to different types of edges:

$$
\begin{aligned}
I(\boldsymbol{x}; \tilde{\boldsymbol{\mu}}_1) &= \sum_i (1 - d_i)\mu_i(x_i) + \sum_{ij \in E} \mu_{ij}(x_i, x_j) \\
&= \sum_{i \in o} (1 - d_i)\tilde{\tau}_1 + \sum_{i \in h} (1 - d_i)\tilde{\mu}_{1,i}(x_i) + \sum_{ij \in E_h} \tilde{\mu}_{1,ij}(x_i, x_j) \\
&\quad + \sum_{ij \in E_{oh}} \tilde{\mu}_{1,j}(x_j) + \sum_{ij \in E_o} \tilde{\tau}_1 \\
&= 0
\end{aligned}
$$

Since the subgraph of observed nodes is a forest, it has $|E_o| = |o| - |P_o|$ edges. Furthermore, $\sum_{i \in o} d_i = |E_{oh}| + 2|E_o|$ so we can rewrite the above expression as:

$$
I(\boldsymbol{x}; \tilde{\boldsymbol{\mu}}_1) = (|P_o| - |E_{oh}|)\tilde{\tau}_1 + \sum_{i \in h}(1 - d_i^h)\tilde{\mu}_{1,i}(x_i) + \sum_{ij \in E_h} \tilde{\mu}_{1,ij}(x_i, x_j).
$$

Notice we also combined the summation over $ij \in E_{oh}$ to that over $i \in h$, changing $d_i$ to $d_i^h$. The entire graph being a tree, it must also hold that $|E_{oh}| = |P_h| + |P_o| - 1$. Plugging this into our expression, we get:

$$
I(\boldsymbol{x}; \tilde{\boldsymbol{\mu}}_1) = I(\boldsymbol{x}_h; \tilde{\boldsymbol{\mu}}_{1,h}) + (1 - |P_h|)\tilde{\tau}_1 = 0.
$$

Now because of the way we set $\tilde{\mu}$, we arrive at:

$$
\frac{I(\boldsymbol{x}; \tilde{\boldsymbol{\mu}}_1)}{\tilde{\tau}_1} = I(\boldsymbol{x}_h; \tilde{\boldsymbol{\mu}}) + (1 - |P_h|)\tilde{\tau} = 0,
$$

which gives Eq. (5.4).

Combining the items above, we see that taking $\tau_\mu$ larger than $\tilde{\tau}_1^{-1}$ and all entries of $\mu_o^{-1}$, gives a feasible solution as required. $\qquad \square$

**Lemma 5.4.** *If* $I(\boldsymbol{x}; \boldsymbol{\mu}) > 0$ *then* $\min_{p \in \mathcal{P}(\boldsymbol{\mu})} p(\boldsymbol{x}_h \mid \boldsymbol{x}_o) = \frac{I(\boldsymbol{x}; \boldsymbol{\mu})}{I(\boldsymbol{x}; \boldsymbol{\mu}) + \max_{p \in \mathcal{P}(\boldsymbol{\mu})} \sum_{\boldsymbol{z}_h \neq \boldsymbol{x}_h} p(\boldsymbol{z}_h, \boldsymbol{x}_o)}.$

*Proof.* Obviously the right hand side is a lower bound on the minimum, we need to show there is a feasible solution that gives this bound. When $I(\boldsymbol{x}; \boldsymbol{\mu}) > 0$ it is easy to see that the conditions of Lem. 5.1 hold. So defining $\tilde{\mu}_1, \tilde{\tau}_1$ as we did in the proof of Lem. 5.3, we can assume $\tilde{\boldsymbol{\mu}}_1 \leq \boldsymbol{\mu} - \mathbf{I}_{\boldsymbol{x}}, I(\boldsymbol{x}_h; \tilde{\boldsymbol{\mu}}_1) + (1 - |P_h|) = 0$. Now consider setting:

$$
\tau_\mu = \frac{1}{I(\boldsymbol{x}, \boldsymbol{\mu}) + \tilde{\tau}_1}, \quad \tilde{\boldsymbol{\mu}} = \tilde{\boldsymbol{\mu}}_{1,h}\tau_\mu, \quad \tilde{\tau} = \tilde{\tau}_1 \tau_\mu, \quad \mu_{\boldsymbol{x}} = I(\boldsymbol{x}; \boldsymbol{\mu})\tau_\mu.
$$

Since $\tilde{\tau}_1$ is defined as the value of the maximization problem in the denominator of the bound stated in the lemma, it can be seen that the value of $\mu_{\boldsymbol{x}}$ is equal to this bound. So if this solution is feasible for Eq. (5.3), $\mu_{\boldsymbol{x}}$ is also an upper bound on the robust conditional probability and it must also be the optimal value. We will simply go through each constraint in Eq. (5.3) and show this solution satisfies it:

- $\tilde{\boldsymbol{\mu}} \in \mathcal{M}_L^h, 0 \leq \tilde{\boldsymbol{\mu}} \leq \tau_\mu \boldsymbol{\mu}_h - \mu_{\boldsymbol{x}} \mathbb{I}_{\boldsymbol{x}_h}$: since $\tilde{\mu}_1 \in \mathcal{M}_L(U)$ and linear constraints stay satisfied after multiplying all variables by a positive scalar, we have $\tilde{\boldsymbol{\mu}} \in \mathcal{M}_L^h$. Satisfaction of capacity constraints is also a direct consequence of $\tilde{\mu}_1$ satisfying capacity constraints: $\tilde{\boldsymbol{\mu}} = \tilde{\boldsymbol{\mu}}_{1,h}\tau_\mu \leq (\boldsymbol{\mu}_h - \mathbf{I}_{\boldsymbol{x}})\tau_\mu = \tau_\mu \boldsymbol{\mu}_h - \mu_{\boldsymbol{x}}\mathbb{I}_{\boldsymbol{x}_h}$.

- $\mu_i(x_i)\tau_\mu \geq 1 \quad \forall i \in o, \mu_{ij}(x_i, x_j)\tau_\mu \geq 1 \quad \forall ij \in E_o$: Notice that $\tilde{\mu}_1$ also has components for observed variables $i \in o$ that satisfy $\tilde{\tau}_1 = \tilde{\mu}_{1,i}(x_i) \leq \mu_i(x_i) - I(\boldsymbol{x}; \boldsymbol{\mu})$ and $\tilde{\tau}_1 = \tilde{\mu}_{1,ij}(x_i, x_j) \leq \mu_{ij}(x_i, x_j) - I(\boldsymbol{x}; \boldsymbol{\mu})$ for $ij \in E_o$. This gives us the constraints easily:

$$
\tilde{\tau}_1 + I(\boldsymbol{x}; \boldsymbol{\mu}) = \frac{1}{\tau_\mu} \leq \mu_i(x_i) \quad \forall i \in o,
$$

and the same holds for every $ij \in E_o$.

- $\sum_{z_i} \tilde{\mu}_i(z_i) = \tilde{\tau} \quad \forall i \in h, \mu_{\boldsymbol{x}} + \tilde{\tau} = 1$: Easy to see from our setting of $\tilde{\boldsymbol{\mu}}, \tilde{\tau}, \mu_{\boldsymbol{x}}$, because $\tilde{\boldsymbol{\mu}}_1$ normalizes to $\tilde{\tau}_1$.

- $I(\boldsymbol{x}_h; \tilde{\boldsymbol{\mu}}) + (1 - |P_h|)\tilde{\tau} \leq 0, \tau_\mu I(\boldsymbol{x}; \boldsymbol{\mu}) - \mu_{\boldsymbol{x}} - I(\boldsymbol{x}_h; \tilde{\boldsymbol{\mu}}) + (|P_h| - 1)\tilde{\tau} \leq 0$: Using $I(\boldsymbol{x}_h; \tilde{\boldsymbol{\mu}}) + (1 - |P_h|)\tilde{\tau} = 0$ (this was proved in the proof of Lem. 5.3) and because we set $\mu_{\boldsymbol{x}} = I(\boldsymbol{x}; \boldsymbol{\mu})\tau_\mu$, it is easy to confirm these two constraints are satisfied.

$\square$

We are left with the task of proving Lem. 5.2 and Lem. 5.1, this is the topic of the next section.

### 5.1 Proofs of Lem. 5.2 and Lem. 5.1

The problem we are concerned with, Eq. (5.1), has an exponential number of constraints. We will see shortly that these constraints can be treated as constraints on the value of 2nd-best MAP problems [5], one over the tree shaped field $\lambda(\boldsymbol{z})$ and the other over the forest shaped $\lambda(\boldsymbol{z}_h, \boldsymbol{x}_o)$. To prove our results we will use a relaxation of these problems. Specifically, we will use the tightness of this relaxation in trees and forests to switch these constraints with a polynomially sized set, that is easier to handle analytically. Hence we turn to derive the set of linear constraints, this is done in a very similar manner to the derivation in [6].

#### 5.1.1 Second Best MAP using Dual Decomposition

As proved by the authors in [5], the 2nd-best MAP problem over a field $\lambda(\boldsymbol{z})$, with excluded assignment $\boldsymbol{x}$ can be written as follows:

$$\max_{\tilde{\boldsymbol{\mu}}} \boldsymbol{\lambda} \cdot \tilde{\boldsymbol{\mu}}$$

$$\text{s.t. } \tilde{\boldsymbol{\mu}} \in \mathcal{M}_L, \tilde{I}(\boldsymbol{x}\,;\,\tilde{\boldsymbol{\mu}}) \leq |P| - 1,$$

where $|P|$ is the number of connected components. This is in fact a relaxation of the 2nd-best MAP problem, but it is exact when the graph is a tree or a forest. The dual of this problem is:

$$\min_{\boldsymbol{\delta}, \delta_{\boldsymbol{x}}} \sum_i \delta_i + \sum_{ij} \delta_{ij} + (|P| - 1)\delta_{\boldsymbol{x}}$$

$$\text{s.t. } \lambda_i(z_i) + \sum_j \delta_{ji}(z_i) + (d_i - 1)\delta_{\boldsymbol{x}} \mathbb{I}_{z_i = x_i} \leq \delta_i \quad \forall i, z_i$$

$$\lambda_{ij}(z_i, z_j) - \delta_{ji}(z_i) - \delta_{ij}(z_j) - \delta_{\boldsymbol{x}} \mathbb{I}_{z_i, z_j = x_i, x_j} \leq \delta_{ij} \quad \forall ij, (z_i, z_j)$$

$$\delta_{\boldsymbol{x}} \geq 0$$

At the optimum, $\delta_i, \delta_{ij}$ will just be equal to the maximum of the left hand side over different values of $z_i, z_j$ (since the problem is a minimization problem), hence we can solve:

$$\min_{\boldsymbol{\delta}, \delta_{\boldsymbol{x}} \geq 0} \sum_i \max_{z_i} \left\{ \lambda_i(z_i) + \sum_j \delta_{ji}(z_i) + (d_i - 1)\delta_{\boldsymbol{x}} \mathbb{I}_{z_i = x_i} \right\} +$$

$$\sum_{ij} \max_{z_i, z_j} \left\{ \lambda_{ij}(z_i, z_j) - \delta_{ji}(z_i) - \delta_{ij}(z_j) - \delta_{\boldsymbol{x}} \mathbb{I}_{z_i, z_j = x_i, x_j} \right\} + (|P| - 1)\delta_{\boldsymbol{x}}$$

To formulate a set of linear constraints that are satisfied if and only if this MAP value is smaller than a constant $c$, we can use auxiliary variables and a polynomial number of constraints, as done in [3]:

$$\sum_i \alpha_i + \sum_{ij} \alpha_{ij} + (|P| - 1)\delta_{\boldsymbol{x}} \leq c \tag{5.5}$$

$$\lambda_i(z_i) + \sum_j \delta_{ji}(z_i) + (d_i - 1)\delta_{\boldsymbol{x}} \mathbb{I}_{z_i = x_i} \leq \alpha_i \quad \forall i, z_i$$

$$\lambda_{ij}(z_i, z_j) - \delta_{ji}(z_i) - \delta_{ij}(z_j) - \delta_{\boldsymbol{x}} \mathbb{I}_{z_i, z_j = x_i, x_j} \leq \alpha_{ij} \quad \forall ij, (z_i, z_j)$$

$$\delta_{\boldsymbol{x}} \geq 0.$$

In the next section we will place these constraints in Eq. (5.1) and move back to its own dual, after some manipulation this will give us Lem. 5.2.

### 5.1.2 Concluding the Proofs

*Proof of Lem. 5.2.* Consider Eq. (5.1). Because we know that the optimal value of $\lambda_{\boldsymbol{x}}$ is in the segment $[0,1]$, this problem can be written as:

$$\max \ \lambda_{\boldsymbol{x}} \tag{5.6}$$
$$\text{s.t.} \ \max_{\boldsymbol{z} \neq \boldsymbol{x}} \lambda(\boldsymbol{z}) \leq 0$$
$$\max_{\boldsymbol{z}_h \neq \boldsymbol{x}_h, \boldsymbol{z}_o = \boldsymbol{x}_o} \lambda(\boldsymbol{z}) \leq -\lambda_{\boldsymbol{x}}$$
$$\lambda(x_r) + \sum_{i \neq r} \lambda_{i,pa(i)}(x_i, x_{pa(i)}) + \lambda_i(x_i) \leq 1 - \lambda_{\boldsymbol{x}}$$
$$\lambda \cdot \boldsymbol{\mu} \geq 0.$$

Begin by writing the full dual problem, where we plug the liner constraints described in Eq. (5.5) instead of the first two constraints in Eq. (5.6). The first $4$ constraints are received by replacing the first 2nd-best MAP in Eq. (5.6), while the $4$ constraints after these are for the second 2nd-best MAP in Eq. (5.6). On the right hand side we assign dual variables to each of the constraints:

$$\max \ \lambda_{\boldsymbol{x}}$$

| | |
|---|---:|
| s.t. $\sum_i \alpha_i + \sum_{ij} \alpha_{ij} \leq 0$ | $\bar{\tau}$ |
| $\lambda_i(z_i) + \sum_j \bar{\delta}_{ji}(z_i) + (d_i - 1)\bar{\delta}_{\boldsymbol{x}} \mathbb{I}_{z_i = x_i} \leq \alpha_i \quad \forall i, z_i$ | $\bar{\mu}_i(z_i)$ |
| $\lambda_{ij}(z_i, z_j) - \bar{\delta}_{ji}(z_i) - \bar{\delta}_{ij}(z_j) - \bar{\delta}_{\boldsymbol{x}} \mathbb{I}_{z_i, z_j = x_i, x_j} \leq \alpha_{ij} \quad \forall ij, (z_i, z_j)$ | $\bar{\mu}_{ij}(z_i, z_j)$ |
| $\bar{\delta}_{\boldsymbol{x}} \geq 0$ | |
| $\sum_{i \in h} \beta_i + \sum_{ij \in E_h} \beta_{ij} + (|P_h| - 1)\tilde{\delta}_{\boldsymbol{x}} \leq -\lambda_{\boldsymbol{x}} - \sum_{ij \in E_o} \lambda_{ij}(x_i, x_j) - \sum_{i \in o} \lambda_i(x_i)$ | $\tilde{\tau}$ |
| $\lambda_i(z_i) + \sum_{j \in o} \lambda_{ji}(x_j, z_i) + \sum_{j \in h} \tilde{\delta}_{ji}(z_i) + (d_i^h - 1)\tilde{\delta}_{\boldsymbol{x}} \mathbb{I}_{z_i = x_i} \leq \beta_i \quad \forall i \in h, z_i$ | $\tilde{\mu}_i(z_i)$ |
| $\lambda_{ij}(z_i, z_j) - \tilde{\delta}_{ji}(z_i) - \tilde{\delta}_{ij}(z_j) - \tilde{\delta}_{\boldsymbol{x}} \mathbb{I}_{z_i, z_j = x_i, x_j} \leq \beta_{ij} \quad \forall ij \in E_h, (z_i, z_j)$ | $\tilde{\mu}_{ij}(z_i, z_j)$ |
| $\tilde{\delta}_{\boldsymbol{x}} \geq 0$ | |
| $\lambda_r(x_r) + \sum_{i \neq r} \lambda_{i,pa(i)}(x_i, x_{pa(i)}) + \lambda_i(x_i) \leq 1 - \lambda_{\boldsymbol{x}}$ | $\mu_{\boldsymbol{x}}$ |
| $\lambda \cdot \boldsymbol{\mu} \geq 0$ | $\tau_\mu$ |

Because we assume $(V, E)$ is connected, the coefficient of $\bar{\delta}_{\boldsymbol{x}}$ in the first constraint is 0 and this variable does not appear in the constraint. Yet the subgraph of hidden variables might not be connected. Recall we denoted its number of connected components by $|P_h|$, this explains the coefficient of $\bar{\delta}_{\boldsymbol{x}}$ in the fifth consraint. Now we take the dual of the above and get the problem:

$$\min \ \mu_{\boldsymbol{x}}$$

| | |
|---|---:|
| s.t. $\mu_{\boldsymbol{x}} + \tilde{\tau} = 1$ | $\lambda_{\boldsymbol{x}}$ |
| $\bar{\mu}_i(z_i) + \tilde{\mu}_i(z_i) - \mu_i(z_i)\tau_\mu + \mathbb{I}_{z_i = x_i}\mu_{\boldsymbol{x}} = 0 \quad \forall i \in h, z_i$ | $\lambda_i(z_i), i \in h$ |
| $\bar{\mu}_{ij}(z_i, z_j) + \tilde{\mu}_{ij}(z_i, z_j) - \mu_{ij}(z_i, z_j)\tau_\mu + \mathbb{I}_{z_i, z_j = x_i, x_j}\mu_{\boldsymbol{x}} = 0 \quad \forall ij \in E_h, (z_i, z_j)$ | $\lambda_{ij}(z_i, z_j)$ |
| $\bar{\mu}_i(z_i) + \mathbb{I}_{z_i = x_i}(\tilde{\tau} + \mu_{\boldsymbol{x}}) - \mu_i(z_i)\tau_\mu = 0 \quad \forall i \in o, z_i$ | $\lambda_i(z_i), i \in o$ |
| $\bar{\mu}_{ij}(z_i, z_j) + \mathbb{I}_{z_i, z_j = x_i, x_j}(\tilde{\tau} + \mu_{\boldsymbol{x}}) - \mu_{ij}(z_i, z_j)\tau_\mu = 0 \quad \forall ij \in E_o, (z_i, z_j)$ | $\lambda_{ij}(z_i, z_j)$ |
| $\bar{\mu}_{ij}(z_i, z_j) + \mathbb{I}_{z_j = x_j}(\tilde{\mu}_i(z_i) + \mathbb{I}_{z_i = x_i}\mu_{\boldsymbol{x}}) - \mu_{ij}(z_i, z_j)\tau_\mu = 0 \quad \forall ij \in E_{ho}, (z_i, z_j)$ | $\lambda_{ij}(z_i, z_j)$ |
| $\sum_{z_j} \bar{\mu}_{ij}(z_i, z_j) = \bar{\mu}_i(z_i) \quad \forall ij \in E, z_i$ | $\bar{\delta}_{ji}(z_i)$ |
| $\sum_{z_j} \tilde{\mu}_{ij}(z_i, z_j) = \tilde{\mu}_i(z_i) \quad \forall ij \in E_h, z_i$ | $\tilde{\delta}_{ji}(z_i)$ |
| $\sum_{z_i} \bar{\mu}_i(z_i) = \bar{\tau} \quad \forall i$ | $\alpha_i$ |
| $\sum_{z_i} \tilde{\mu}_i(z_i) = \tilde{\tau} \quad \forall i$ | $\beta_i$ |
| $\sum_i (1 - d_i)\bar{\mu}_i(x_i) + \sum_{ij} \bar{\mu}_{ij}(x_i, x_j) \leq 0$ | $\bar{\delta}_{\boldsymbol{x}}$ |
| $\sum_i (1 - d_i^h)\tilde{\mu}_i(x_i) + \sum_{ij} \tilde{\mu}_{ij}(x_i, x_j) + (1 - |P_h|)\tilde{\tau} \leq 0$ | $\tilde{\delta}_{\boldsymbol{x}}$ |

All variables in the problem are constrained to be non negative as well. The right column denotes the primal variables that each dual constraint corresponds to, in the third row these variables are $\lambda_{ij}$ for $ij \in E_h$, while in the fifth and sixth they are for $ij \in E_o$ and $E_{ho}$ respectively. Notice that we can simplify the problem by using the second to sixth equality constraints and eliminate variables $\bar{\mu}$. Local consistency constraints for $\bar{\mu}$:

$$\sum_{z_j} \bar{\mu}_{ij}(z_i, z_j) = \bar{\mu}_i(z_i) \quad \forall ij \in E, z_i,$$

will be satisfied because of $\tilde{\mu}$ and $\mu$'s local consistency, while normalization constraints:

$$\sum_{z_i} \bar{\mu}_i(z_i) = \bar{\tau} \quad \forall i,$$

are also satisfied because $\tilde{\mu}$ normalizes to $\tilde{\tau}$. Combining the above switch of variables into the constraint $\tilde{I}(\boldsymbol{x}\,;\,\bar{\mu}) \le 0$, it becomes:

$$\tau_\mu \tilde{I}(\boldsymbol{x}\,;\,\boldsymbol{\mu}) - \mu_{\boldsymbol{x}} - \tilde{I}(\boldsymbol{x}_h\,;\,\tilde{\boldsymbol{\mu}}) + (\sum_{i \in o}(d_i - 1) - |E_o|)\tilde{\tau} \le 0.$$

We already showed in the proof of Lem. 5.3 that the term $\sum_{i \in o}(d_i - 1) - |E_o|$ is equal to $|P_h| - 1$, turning the above constraint to:

$$\tau_\mu \tilde{I}(\boldsymbol{x}\,;\,\boldsymbol{\mu}) - \mu_{\boldsymbol{x}} - \tilde{I}(\boldsymbol{x}_h\,;\,\tilde{\boldsymbol{\mu}}) + (|P_h| - 1)\tilde{\tau} \le 0.$$

So we end up with the following problem:

$$\min \mu_{\boldsymbol{x}}$$

$$\begin{aligned}
\text{s.t. } & \mu_{\boldsymbol{x}} + \tilde{\tau} = 1 \\
& \tilde{\mu}_i(z_i) - \mu_i(z_i)\tau_\mu + \mathbb{I}_{z_i = x_i}\mu_{\boldsymbol{x}} \le 0 \quad \forall i \in h, z_i \\
& \tilde{\mu}_{ij}(z_i, z_j) - \mu_{ij}(z_i, z_j)\tau_\mu + \mathbb{I}_{z_i, z_j = x_i, x_j}\mu_{\boldsymbol{x}} \le 0 \quad \forall ij \in E_h, (z_i, z_j) \\
& \mu_i(x_i)\tau_\mu \ge 1 \quad \forall i \in o \\
& \mu_{ij}(x_i, x_j)\tau_\mu \ge 1 \quad \forall ij \in E_o \\
& \tilde{\mu}_i(z_i) + \mathbb{I}_{z_i = x_i}\mu_{\boldsymbol{x}} - \mu_{ij}(z_i, x_j)\tau_\mu \le 0 \quad \forall ij \in E_{ho} \\
& \sum_{z_j} \tilde{\mu}_{ij}(z_i, z_j) = \tilde{\mu}_i(z_i) \quad \forall ij \in E_h, z_i \\
& \sum_{z_i} \tilde{\mu}_i(z_i) = \tilde{\tau} \quad \forall i \in h \\
& \tau_\mu I(\boldsymbol{x}\,;\,\boldsymbol{\mu}) - \mu_{\boldsymbol{x}} - I(\boldsymbol{x}_h\,;\,\tilde{\boldsymbol{\mu}}) + (|P_h| - 1)\tilde{\tau} \le 0 \\
& I(\boldsymbol{x}_h\,;\,\tilde{\boldsymbol{\mu}}) + (1 - |P_h|)\tilde{\tau} \le 0
\end{aligned}$$

Simplifying notation using the vectors $\boldsymbol{\mu}_h, \mathbb{I}_{\boldsymbol{x}}, \boldsymbol{\mu}_o$ that we defined in Section 2, the problem takes the shape of Eq. (5.3) $\qquad\square$

*Proof of Lem. 5.2.* From Thm. 4.2 we know that:

$$\max_{\tilde{\boldsymbol{\mu}} \in \mathcal{M}_L(U), \tilde{\boldsymbol{\mu}} \le \boldsymbol{\mu}} Z(\tilde{\boldsymbol{\mu}}) = \max_{p \in \mathcal{P}(\boldsymbol{\mu})} \sum_{\boldsymbol{u} \in U} p(\boldsymbol{u}),$$

$$\max_{\substack{\tilde{\boldsymbol{\mu}} \in \mathcal{M}_L(U), \tilde{\boldsymbol{\mu}} \le \boldsymbol{\mu} \\ I(\boldsymbol{x}\,;\,\tilde{\boldsymbol{\mu}}) \le 0}} Z(\tilde{\boldsymbol{\mu}}) = \max_{p \in \mathcal{P}(\boldsymbol{\mu})} \sum_{\boldsymbol{u} \in U \setminus \boldsymbol{x}} p(\boldsymbol{u}).$$

Now for each $i, (i, j) \in E$, consider replacing constraints in $\mathcal{P}(\boldsymbol{\mu})$ as follows:

$$\sum_{\boldsymbol{z}:z_i, z_j = x_i, x_j} p(\boldsymbol{z}) = \mu_{ij}(x_i, x_j) \rightarrow \sum_{\substack{\boldsymbol{z}:z_i, z_j = x_i, x_j, \\ \boldsymbol{z} \ne \boldsymbol{x}}} p(\boldsymbol{z}) \le \mu_{ij}(x_i, x_j) - I(\boldsymbol{x}, \boldsymbol{\mu}),$$

$$\sum_{\boldsymbol{z}:z_i = x_i} p(\boldsymbol{z}) = \mu_i(x_i) \rightarrow \sum_{\substack{\boldsymbol{z}:z_i = x_i \\ \boldsymbol{z} \ne \boldsymbol{x}}} p(\boldsymbol{z}) \le \mu_i(x_i) - I(\boldsymbol{x}, \boldsymbol{\mu}).$$

We will denote this set by $\tilde{\mathcal{P}}(\boldsymbol{\mu})$. Since for any $p \in \mathcal{P}(\boldsymbol{\mu})$ we know that $p(\boldsymbol{x}) \ge I(\boldsymbol{x}, \boldsymbol{\mu})$, it holds that $\mathcal{P}(\boldsymbol{\mu}) \subseteq \tilde{\mathcal{P}}(\boldsymbol{\mu})$, which means the maximum of the new problem is *higher* than that of the original for both problems (on $U$ and $U \setminus \boldsymbol{x}$):

$$\max_{p \in \mathcal{P}(\boldsymbol{\mu})} \sum_{\boldsymbol{u} \in U} p(\boldsymbol{u}) \le \max_{p \in \tilde{\mathcal{P}}(\boldsymbol{\mu})} \sum_{\boldsymbol{u} \in U} p(\boldsymbol{u})$$

$$\max_{p \in \mathcal{P}(\boldsymbol{\mu})} \sum_{\boldsymbol{u} \in U \setminus \boldsymbol{x}} p(\boldsymbol{u}) \le \max_{p \in \tilde{\mathcal{P}}(\boldsymbol{\mu})} \sum_{\boldsymbol{u} \in U \setminus \boldsymbol{x}} p(\boldsymbol{u})$$

Taking the dual of this new problem on $U \setminus \boldsymbol{x}$ we obtain:

$$\min_{\boldsymbol{\lambda}} \; \boldsymbol{\lambda} \cdot (\boldsymbol{\mu} - \mathbf{I}_{\boldsymbol{x}})$$

$$\text{s.t. } \lambda(\boldsymbol{z}) \geq 1 \qquad\qquad\qquad \forall \boldsymbol{z} \in U \setminus \boldsymbol{x}$$
$$\lambda(\boldsymbol{z}) \geq 0 \qquad\qquad\qquad \forall \boldsymbol{z} \notin U$$
$$\lambda_{ij}(x_i, x_j) \geq 0, \lambda_i(x_i) \geq 0 \qquad \forall i \in V, (i,j) \in E$$

From the result in Cor. 3.1, we can consider the variables to be non-negative (i.e. $\boldsymbol{\lambda} \geq 0$), the second constraint is redundant and can be removed. Furthermore, the first constraint is in fact a constraint on the value of the 2nd-best MAP problem on $-\lambda(\boldsymbol{z})$ (i.e. minimization of $\lambda(\boldsymbol{z})$ while excluding $\boldsymbol{x}$). Adapting the constraints in Eq. (5.5) to a minimization problem and switching into our problem we get:

$$\min_{\boldsymbol{\lambda} \geq 0, \delta_{\boldsymbol{x}} \geq 0, \boldsymbol{\alpha}, \boldsymbol{\delta}} \boldsymbol{\lambda} \cdot (\boldsymbol{\mu} - \mathbf{I}_{\boldsymbol{x}}) \qquad\qquad (5.7)$$

$$\text{s.t. } \sum_i \alpha_i + \sum_{ij} \alpha_{ij} \geq 1$$

$$\lambda_i(z_i) + \sum_j \delta_{ji}(z_i) + (1 - d_i)\delta_{\boldsymbol{x}}\mathbb{I}_{z_i = x_i} \geq \alpha_i \quad \forall i, z_i \in \bar{X}_i$$

$$\lambda_{ij}(z_i, z_j) - \delta_{ji}(z_i) - \delta_{ij}(z_j) + \delta_{\boldsymbol{x}}\mathbb{I}_{z_i, z_j = x_i, x_j} \geq \alpha_{ij} \quad \forall ij, (z_i, z_j) \in \bar{X}_i \times \bar{X}_j.$$

Taking the dual of this problem, it is easy to see it equals to:

$$\max_{\substack{\tilde{\boldsymbol{\mu}} \in \mathcal{M}_L(U), \tilde{\boldsymbol{\mu}} \leq \boldsymbol{\mu} - \mathbf{I}_{\boldsymbol{x}} \\ I(\boldsymbol{x}\,;\tilde{\boldsymbol{\mu}}) \leq 0}} Z(\tilde{\boldsymbol{\mu}}).$$

The constraints of this problem are more strict than the ones in the original, therefore its value is *lower*:

$$\max_{p \in \mathcal{P}(\boldsymbol{\mu})} \sum_{\boldsymbol{u} \in U \setminus \boldsymbol{x}} p(\boldsymbol{u}) = \max_{\substack{\tilde{\boldsymbol{\mu}} \in \mathcal{M}_L(U), \tilde{\boldsymbol{\mu}} \leq \boldsymbol{\mu} \\ I(\boldsymbol{x}\,;\tilde{\boldsymbol{\mu}}) \leq 0}} Z(\tilde{\boldsymbol{\mu}}) \geq \max_{\substack{\tilde{\boldsymbol{\mu}} \in \mathcal{M}_L(U), \tilde{\boldsymbol{\mu}} \leq \boldsymbol{\mu} - \mathbf{I}_{\boldsymbol{x}} \\ I(\boldsymbol{x}\,;\tilde{\boldsymbol{\mu}}) \leq 0}} Z(\tilde{\boldsymbol{\mu}}) = \max_{p \in \tilde{\mathcal{P}}(\boldsymbol{\mu})} \sum_{\boldsymbol{u} \in U \setminus \boldsymbol{x}} p(\boldsymbol{u}).$$

We gather that an equality must hold:

$$\max_{p \in \mathcal{P}(\boldsymbol{\mu})} \sum_{\boldsymbol{u} \in U \setminus \boldsymbol{x}} p(\boldsymbol{u}) = \max_{p \in \tilde{\mathcal{P}}(\boldsymbol{\mu})} \sum_{\boldsymbol{u} \in U \setminus \boldsymbol{x}} p(\boldsymbol{u}) = \max_{\substack{\tilde{\boldsymbol{\mu}} \in \mathcal{M}_L(U), \tilde{\boldsymbol{\mu}} \leq \boldsymbol{\mu} - \mathbf{I}_{\boldsymbol{x}} \\ I(\boldsymbol{x}\,;\tilde{\boldsymbol{\mu}}) \leq 0}} Z(\tilde{\boldsymbol{\mu}}).$$

To complete the proof we need to show the existence a solution $\tilde{\boldsymbol{\mu}}$ that is optimal for the problem on the right hand side and satisfies $I(\boldsymbol{x}\,;\tilde{\boldsymbol{\mu}}) = 0$. Then assume towards contradiction that Eq. (5.2) holds and there is no optimal solution where $I(\boldsymbol{x}\,;\tilde{\boldsymbol{\mu}}) = 0$. Since the problem is feasible, some optimal solution $\boldsymbol{\mu}^*$ does exist and from complementary slackness, there is a corresponding solution $\boldsymbol{\lambda}^*, 0, \boldsymbol{\alpha}^*, \boldsymbol{\delta}^*$ to Eq. (5.7). Since the value of $\delta_{\boldsymbol{x}}$ is 0, then $\boldsymbol{\lambda}^*, \boldsymbol{\alpha}^*, \boldsymbol{\delta}^*$ is also a feasible solution to the dual of:

$$\max_{p \in \tilde{\mathcal{P}}(\boldsymbol{\mu})} \sum_{\boldsymbol{u} \in U} p(\boldsymbol{u}),$$

which means $\boldsymbol{\lambda}^* \cdot (\boldsymbol{\mu} - \mathbf{I}_{\boldsymbol{x}})$ is an upper bound on this problem. To conclude, we concatenate the inequalities we have so far:

$$\max_{p \in \mathcal{P}(\boldsymbol{\mu})} \sum_{\boldsymbol{u} \in U} p(\boldsymbol{u}) \leq \max_{p \in \tilde{\mathcal{P}}(\boldsymbol{\mu})} \sum_{\boldsymbol{u} \in U} p(\boldsymbol{u}) \leq \boldsymbol{\lambda}^* \cdot (\boldsymbol{\mu} - \mathbf{I}_{\boldsymbol{x}}) = \max_{p \in \mathcal{P}(\boldsymbol{\mu})} \sum_{\boldsymbol{u} \in U \setminus \boldsymbol{x}} p(\boldsymbol{u}).$$

This inequality contradicts the hard inequality we assumed at the statement of the lemma, therefore there exists an optimal solution where $I(\boldsymbol{x}\,;\tilde{\boldsymbol{\mu}}) = 0$ and we can incorporate this equality into the constraints without changing the value of the problem. □