[Reviews · NeurIPS 2017]

Reviewer 1



This paper considers the problem of bounding conditional probabilities given second-order marginals of the features. The bounds are used in a semi-supervised learning setting, and shown to be comparable with VAE, but not as good as Ladder. The problem of estimating lower bounds of conditional probabilities is interesting. While the approach of estimating conditional probabilities using parametric models can lead to estimation errors due to the parametric presentation, the lower bounds can introduce errors due to being overly conservative and seems to require approximations in practice (section 6). It would be helpful to comment on this. The writing could have been improved. For example, in Section 6, it is unclear how continuous features are turned to discrete ones. Does MLP stand for multi-layer perceptron? Does Fig 1 report errors (instead of accuracies)? How are the benchmark algorithms set up? In the caption for Fig 2, red should be green?

Reviewer 2



This paper studies the problem of computing probability bounds, more specifically bounds over probability of atoms of the joint space and conditional probabilities of the class, under the assumption that only some pairwise marginal as well as some univariate marginal values are known. The idea is that such marginals may be easier to obtain than fully specified probabilities, and that cautious inferences can then be used to produce predictions. It is shown that when the marginals follow a tree structure (results are extended to a few other structures), then the problem can actually be solved in closed, analytical form, relating it to cover set and maximum flow problems. Some experiments performed on neural networks show that this simple method is actually competitive with other more complex approaches (Ladder, VAE), while outperforming methods of comparable complexity. The paper is elegantly written, with quite understandable and significant results. I see no reason to not accept it. Authors may be interested in looking at the following papers coming from the imprecise probability literature (since they deal with partially specified probabilities, this may be related): * Benavoli, A., Facchini, A., Piga, D., & Zaffalon, M. (2017). SOS for bounded rationality. arXiv preprint arXiv:1705.02663. * Miranda, E., De Cooman, G., & Quaeghebeur, E. (2007). The Hausdorff moment problem under finite additivity. Journal of Theoretical Probability, 20(3), 663-693. Typos: * L112: solved Our —> solved. Our * L156: assume are —> assume we are * References: 28/29 are redundant

Reviewer 3



The paper introduces results on bounding the joint and conditional probability of multivariate probability distributions, given knowledge of univariate and bivariate marginals. Theorem 4.1 is to my opinion a novel and interesting result, which defines the core of the paper. In addition, solutions are provided to interpret the optimization problem in this theorem. Two applications, one in semi-supervised learning and another in multi-label classification, are studied. Despite being dense and expecting a lot of background from the reader, I think that this paper is well written. At least I was able to understand the theoretical results after doing some effort. Moreover, I also believe that the results might be useful for different types of applications in machine learning. Apart from the applications mentioned in the paper, I believe that the results could also be used to provide tighter worst-case regret bounds in multi-label classification when using the Hamming loss as a surrogate for the subset zero-one loss. In that scenario, optimizing the Hamming loss corresponds to marginals, whereas subset zero-one loss corresponds to the joint distribution, see. e.g. Dembczysnki et al. On label dependence and loss minimization in multi-label classification. Machine Learning 2012. To my opinion, this is an interesting paper.